# Renin-Angiotensin System in Lung Tumor and Microenvironment Interactions

**DOI:** 10.3390/cancers12061457

**Published:** 2020-06-03

**Authors:** Maria Joana Catarata, Ricardo Ribeiro, Maria José Oliveira, Carlos Robalo Cordeiro, Rui Medeiros

**Affiliations:** 1Tumour & Microenvironment Interactions Group, I3S-Institute for Health Research & Innovation, University of Porto, 4200-135 Porto, Portugal; ricardo.ribeiro@i3s.up.pt (R.R.); mariajo@ineb.up.pt (M.J.O.); 2Tumour & Microenvironment Interactions Group, INEB-Biomedical Engineering Institute, University of Porto, 4200-135 Porto, Portugal; 3Department of Pulmonology, Centro Hospitalar e Universitário de Coimbra, 3000-075 Coimbra, Portugal; carlos.crobalo@gmail.com; 4Faculty of Medicine, University of Porto, 4200-319 Porto, Portugal; ruimedei@ipoporto.min-saude.pt; 5Molecular Oncology and Viral Pathology Group-Research Centre, Portuguese Institute of Oncology, 4200-072 Porto, Portugal; 6Laboratory of Genetics, Faculty of Medicine, University of Lisbon, 1649-028 Lisboa, Portugal; 7Department of Clinical Pathology, Centro Hospitalar e Universitário de Coimbra, 3000-075 Coimbra, Portugal; 8Faculty of Medicine, University of Coimbra, 3000-548 Coimbra, Portugal

**Keywords:** lung cancer, renin-angiotensin, hypoxia, tumor microenvironment

## Abstract

The mechanistic involvement of the renin-angiotensin system (RAS) reaches beyond cardiovascular physiopathology. Recent knowledge pinpoints a pleiotropic role for this system, particularly in the lung, and mainly through locally regulated alternative molecules and secondary pathways. Angiotensin peptides play a role in cell proliferation, immunoinflammatory response, hypoxia and angiogenesis, which are critical biological processes in lung cancer. This manuscript reviews the literature supporting a role for the renin-angiotensin system in the lung tumor microenvironment and discusses whether blockade of this pathway in clinical settings may serve as an adjuvant therapy in lung cancer.

## 1. Introduction

Lung cancer is one of the most common malignancies and the most frequent cause of cancer deaths in the past few decades, with over one million people diagnosed worldwide each year [1,2]. Notably, the five-year survival rate is the lowest compared with other frequent malignancies [3]. Its poor prognosis fosters the need for research, namely understanding its aggressiveness mechanisms, in order to improve prevention and treatment.

The renin-angiotensin-aldosterone system is an established primary regulator of blood pressure, homeostasis, and natriuresis [4]. Nevertheless, recent evidence indicates that angiotensin peptides also might have a role in tumor cell proliferation, inflammation and hypoxia, which have been recognized as hallmarks of cancer, actively participating in lung tumor microenvironment regulation [5]. Here, we review current knowledge on the renin-angiotensin system (RAS), considering its potential modulatory role on cancer cell phenotype and on immune surveillance, while exploring the mechanistic association with lung cancer.

## 2. The Renin-Angiotensin System: An Intricate Regulatory Mechanism in Lung Disease Physiopathology?

The RAS was first discovered and studied in the physiological regulation of blood pressure, fluid homeostasis and pathogenesis of hypertension [6]. Angiotensin II (Ang II), the final effector of the system, causes vasoconstriction, both directly and indirectly, by stimulating Ang II type 1 receptor (AT1 receptor) present on the vasculature, and by increasing sympathetic tone and arginine vasopressin release. Chronically, Ang II regulates blood pressure by modulating renal sodium and water reabsorption directly, by stimulating AT1 receptors in the kidney, or indirectly, by stimulating the production and release of aldosterone from the adrenal glands, or stimulating the sensation of thirst in the central nervous system [7]. The enzymatic cascade by which Ang II is produced consists of renin, which cleaves angiotensinogen (AGT) to form the decapeptide angiotensin I (Ang I). Ang I is then further cleaved by angiotensin-converting enzyme (ACE) to produce the octapeptide Ang II, the physiologically active component of the system. Further degradation by aminopeptidase A and N produces angiotensin III (Ang 2–8), and angiotensin IV (Ang 3–8), respectively [7] (Figure 1). The actions of Ang II result from its binding to specific receptors (AT1 and AT2). Ang II, via its AT1 receptor, is also involved in cell proliferation, left ventricular hypertrophy, nephrosclerosis, vascular media hypertrophy, endothelial dysfunction, neointima formation, and processes leading to atherothrombosis [8]. Ang II, via its AT2 receptor, is involved in cell differentiation, tissue repair, apoptosis and vasodilation [8]. Other receptors have been described in relation to the RAS. For instance, an AT4 receptor binding site has been identified, and in contrast to the other AT receptors, it does not seem to be a G protein-coupled receptor [9]. This receptor binds Ang IV (3–8) preferentially, has been localized in various mammalian tissues, and suggested to cause vasodilatation [9].

Recently, a role for local tissue-based RAS has been described, both as modulator of inflammation and of the injury/repair response [10]. In addition, supportive evidence claims that tissue RAS is capable of working synergistically or independently of the systemic RAS, and locally generates angiotensin peptides with regulatory homeostatic functions, thus contributing to dysfunction and/or disease [11] (Figure 1). Within the RAS, the generation of angiotensin peptides is controlled by complex interactions: renin, which is released from the juxtaglomerular apparatus of the kidney, converts the angiotensinogen released by the liver to form angiotensin I. In fact, the liver is the primary source of circulating AGT, a minor but significant fraction of which is filtered through the glomerulus and reabsorbed by proximal tubule cells. It was shown that liver-specific angiotensinogen knockout mice had nearly abolished plasma and renal angiotensinogen protein and renal tissue Ang II [12]. On its turn, the ACE plays a central role in generating Ang II from Ang I [13]. The capillary blood vessels in the lung are one of the major sites of ACE expression and Ang II production in the human body [14]. Alterations in RAS expression were shown to be involved in multiple lung diseases: idiopathic pulmonary fibrosis [15], sarcoidosis [16], pulmonary hypertension [17], acute respiratory distress syndrome [18], and lung cancer [19]. Understanding the involvement of intra-pulmonary RAS on inflammation or fibrosis may open new therapeutic possibilities for the treatment of respiratory diseases. In fact, the increased ACE expression observed in several interstitial lung diseases supports the existence of a pulmonary RAS and establishes a putative role for Ang II in the response to lung injury and fibrosis [10]. Concurringly, findings from a substantial number of in vivo studies demonstrated that ACE inhibitors were capable of attenuating experimental pulmonary fibrosis [20]. Captopril, enalapril, lisinopril and perindopril [21,22,23], could block the effects of bleomycin-, γ irradiation-, amiodarone- and paraquat- induced pulmonary fibrosis in rats. A recent post hoc analysis of data from the placebo arms of phase 3 trials including patients with idiopathic pulmonary fibrosis, showed a slower disease progression in patients under therapy with ACE inhibitors [24]. Therefore, local upregulation of the expression of either Ang II or TGF-β1, which might influence each other’s activity or act in synergy, should be blocked to delay the progression of lung fibrotic disease.

Ang I can also be converted to Ang II through the action of chymase, a chymotrypsin-like enzyme that is expressed in the secretory granules of mast cells [25]. In fact, chymase is the most efficient Ang II-forming enzyme in the human body and has been implicated in a wide variety of human diseases that also implicate its many other protease actions: (1) cleavage of matricellular proteins and peptides, including laminin and fibronectin important for cell survival; (2) activation of peptide and enzyme precursors, including matrix metalloproteinases (MMPs), such as MMP-9, TGF-β, stem cell factor, kallikrein, IL-6 and IL-1β, preproendothelin I and IL-18 [26]. Chymase activity was noticeably increased in the lungs of animals following bleomycin instillation [27]. A recent report showed the prominent accumulation of chymase-expressing mast cells in idiopathic pulmonary arterial hypertension and idiopathic pulmonary fibrosis lungs, and their presence was near the regions with marked TGF-β1 expression [27]. Furthermore, tumor sections from 32 cases of adenocarcinoma and 13 cases of squamous cell carcinoma revealed significantly higher counts of chymase-positive mast cells in the central region than those in the normal regions [28]. Chymase has been associated with TGF-β activation and endothelin-1 formation from preproendothelin-1 in pulmonary fibrosis and chronic obstructive pulmonary disease [26], and in proliferation, adhesion, regulation of E-cadherin expression, and modulation of immunosuppressive microenvironment in lung carcinoma [29,30]. This effect is likely due to TGF-β inhibitory effect on T-cell proliferation and effector activity. Moreover, TGF-β subverts T cell immunity by favoring regulatory T-cell differentiation, further reinforcing immunosuppression at the tumor microenvironment [31].

Ang II is a ligand for two classes of seven trans-membrane G protein-coupled receptors, angiotensin II type 1 (AT1) and angiotensin II type 2 (AT2) receptors [25]. The cellular distribution of AT1 and AT2 receptors in the normal and pathological human lung has been previously described [32]. Whereas *AGTR1* transcript and encoded protein, the AT1 receptor, localized at vascular smooth muscle cells, alveolar macrophages and in the stroma underneath the airway epithelium, *AGTR2* expression, at both mRNA and protein levels, of AT2 receptors were found in bronchial epithelia and endothelial cells [33]. The components of RAS are frequently differentially expressed in various cancers including brain, lung, pancreatic, breast, prostate, colon, skin and cervical carcinomas in comparison with their corresponding non-malignant tissue [34]. A large-scale study of estrogen receptor-positive breast cancer tumors revealed an increase on *AGTR1* mRNA expression. Contradictory results were observed in lung adenocarcinoma tissues since *AGTR1*, *AGTR2* as well as ACE mRNA expression, were underexpressed. Notably, AGT was overexpressed in lung adenocarcinoma tissue [35]. Notably, several physiological and pathophysiological effects of Ang II are mediated mainly through the AT1 receptor [36]. Upon its activation, a pleiotropic regulatory role is induced at the lung, resulting in vasoconstriction, cell proliferation, angiogenesis, and augmentation of pro-inflammatory cytokine production, inflammatory cell chemotaxis, epithelial cell apoptosis, increased oxidative stress and fibrosis [37]. Several in vitro studies demonstrated that the epithelial to mesenchymal transition (EMT) induced by TGF-β1 resulted in an increase in angiotensinogen and AT1 receptor expression, in human lung fibroblasts [38,39,40]. Moreover, active TGF-β1 expression by human lung myofibroblasts was downregulated by AT1 receptor blockade and accompanied the inhibition of collagen synthesis [20]. In contrast, the activation of the AT2 receptor induces opposite effects, although some proinflammatory responses are mediated through the activation of the Nuclear Factor-kB (NF-kB) pathway [37].

Ang II can also be converted into Angiotensin-(1–7) (Ang (1–7)) by the enzymatic activity of angiotensin converting enzyme type 2 (ACE2) or from Ang I, via Angiotensin (1–9) (Ang-(1–9)), a pathway that involves both ACE2 and ACE [34,41] (Figure 1). In the human lung, ACE2 is expressed in endothelial and smooth muscle cells of large and small blood vessels, as well as in alveolar epithelial cells type I and II, and bronchial epithelial cells [42]. ACE2 has a multiplicity of physiological roles that revolve around its trivalent function: a negative regulator of the RAS, facilitator of amino acid transport, and is the entry receptor for SARS-CoV and SARS-CoV-2 [43].

Alternatively, Ang-(1–7) might be produced through hydroxylation of Ang I by the enzyme neprilysin [44]. Neprilysin is a cell membrane-bound zinc metalloprotease that catalyzes the degradation of a number of endogenous vasodilator peptides, such as atrial natriuretic peptide, brain natriuretic peptide, and C-type natriuretic peptide [45]. Physiological actions of neprilysin depend on the balance of its action on the breakdown of vasodilators and vasoconstrictors [46]. Neprilysin has been involved in malignancies, such as prostate, renal and lung cancer [47]. Ang-(1–7) binds to the G protein-coupled receptor Mas, counteracting cardiovascular and baroreflex effects of Ang II through AT1 receptor, and has been suggested to have vasodilator properties and to act as an inhibitor of cell proliferation [48]. It was reported that ACE2 upregulation can reduce lung injury, whereas ACE2 or Ang–(1–7) have a role in the prevention of acute respiratory distress [49,50,51]. ACE2 has recently been identified as the SARS-CoV-2 receptor, the infective agent responsible for COVID-19, providing a critical link between immunity and inflammation [43]. Furthermore, some studies reported that the ACE2/Ang-(1–7)/Mas axis counteracted the ACE/Ang II/AT1 axis in different models of cancer, including lung cancer [52,53,54].

The circulating RAS might also be important in providing the substrates for local modulation of angiotensin peptides with influence in lung diseases. Genetic and epidemiological evidences provide support that germline mutations of RAS components contribute to the risk of developing idiopathic pulmonary fibrosis, acute lung injury and certain malignancies. For example, the human gene encoding angiotensinogen (*AGT*) is located on chromosome 1q42.3, where a large number of single-nucleotide polymorphisms (SNPs) has been described. Among them, the SNPG-6A nucleotide substitution at position 6, upstream from the initial transcription start codon, has been studied particularly. The A allele has been associated in vitro with an increased expression of the *AGT* gene and with higher AGT synthesis [55]. A previous report showed that the G-6A polymorphism of the *AGT* gene is associated with increased angiotensin production and idiopathic pulmonary fibrosis progression [55]. The *AGT* M235T variant conferred a risk for developing breast cancer in post-menopausal women [56]. Moreover, ACE plasma levels depend on a 287 bp insertion/deletion (I/D) polymorphism of the ACE gene located on chromosome 17q23. Homozygotes for the D allele have the highest ACE plasma levels, while homozygotes for the I allele have the lowest, and ID heterozygotes have intermediate levels. Several studies found an association between ACE I/D polymorphism and lung cancer, albeit some conflicting results [56,57,58].

Thus, the altered regulation of RAS homeostasis, either genetically or environmentally-driven, might contribute to the deregulation of intersected axes, thereby modulating lung cancer aggressiveness.

## 3. Renin-Angiotensin System and the Hallmarks of Cancer: Application to Lung Tumors

The RAS is an important signaling system within the tumor microenvironment, promoting both malignant and stromal cell proliferation [5]. Notably, it is also involved in the regulation of metabolic homeostasis and of tumor-associated phenotypes, such as angiogenesis, migration and invasion [5,11,59] (Figure 2).

### 3.1. Cell Proliferation, Invasion and Migration

#### 3.1.1. ACE/Ang II/AT1 Receptor Axis

Ang II is an important growth promoter in a variety of cell types, where it activates the phosphatidylinositol signaling via AT1 receptor and increases cytosolic Ca^2+^, a process linked to mitogenesis [60]. Another intracellular mechanism by which Ang II controls cell growth is through the modulation of protein tyrosine kinases (PTK) [61]. These proteins are coupled to several growth factor receptors, particularly epidermal growth factor receptor (EGFR), and transduce growth factor-induced signals through protein kinase C (PKC), or through MMPs-mediated mechanisms, which include a disintegrin and metalloproteinase (ADAM) family members. This crosstalk was shown to contribute towards cancer [5] (Figure 2). Ang II was shown to stimulate the migration of fibroblasts through activation of AT1 receptor in an EGFR activation-dependent process and be mediated by shedding of heparin-binding-EGF [62]. Noteworthily, previous experiments have shown that EGFR inhibition partially blocked Ang II-induced proliferation of breast intraductal carcinoma cells [63]. Although the exact molecular mechanism underlying the transactivation of EGFR through Ang II-AT1 receptor binding in lung tumor cells is not completely understood, previous clinical studies evidenced an effect of ACE inhibitors or angiotensin type 1 receptor (ATR1) blockers on lung cancer patients, in combination with chemotherapeutics and EGFR inhibitors [64,65].

c-Fos and c-jun genes are members of the activator protein-1 (AP-1) transcription factor family and are well known targets of the ERK1/2 mitogen activated protein kinase (MAPK) pathway [66]. Ang II increased the phosphorylation of the extracellular signal-regulated protein kinase 1/2, c-jun-N-terminal kinase 1/2, or p38 mitogen-activated protein kinases, signaling pathways involved on cell proliferation and particularly relevant for cancer development [67,68] (Figure 2). Notably, in fibroblasts, Ang II binding to AT1R results in Smad2 and Smad3 phosphorylation via the ERK/p38/MAPK pathway, increasing the activation of the TGFβ1/Smad2/Smad3/Smad4 signaling [69]. Telmisartan, an AT1 receptor antagonist, significantly inhibited the growth of the non-small cell lung cancer (NSCLC) A549 cell line, in a time- and dose-dependent manner and led to an increase of the pro-apoptotic proteins caspase-3 and Bcl-xL [70]. Attoub S and colleagues demonstrated that captopril significantly inhibited tumor growth using a non-small cell lung carcinoma LNM35 pre-clinical model [71].

The establishment of distant metastatic disease is associated with poor prognosis in patients with lung cancer. Tumor metastasis involves several biological steps, such as cell-to-cell and cell-to-extracellular matrix (ECM) interactions, degradation of ECM components, and new vessels formation [72], and some studies suggest that cancer stem cells (CSC) contribute, directly and indirectly, to the generation of metastases [73,74,75]. Ang II/ATR1 pathway activation has been shown to promote EMT in cancer cells, which is a hallmark of cancer aggressiveness [76], while chymase, a protease that converts Ang I to Ang II, increased the expression of MMP-9 in a dose dependent manner in two lung cancer cell lines (A549 and H520) [30]. Moreover, it was shown that Ang II induced CSC marker expression and facilitated tumor cell survival in NSCLC cells line [77]. Recent data from a breast cancer model revealed an additional mechanism whereby ATR1 promotes tumor cell contraction, migration and invasion by upregulating the C-X-C chemokine receptor type 4 (CXCR4)/Stromal cell derived factor-1α (SDF-1α) signaling through the Focal adhesion kinase (FAK)/Ras homolog family member A (RhoA)/Rho associated kinase 1/2 (ROCK1/2)/Myosin light-chain kinase (MLC) pathway, ultimately inducing metastatic disease [78]. Taken together, these findings support a role for mediators of the Ang II/ATR1 axis as lung cancer modulators, particularly influencing malignant cell growth, dedifferentiation, and migration (Figure 2).

#### 3.1.2. AT2 Receptor

In contrast to the pro-tumor effects of AT1 receptor signaling, activation of the AT2 receptor has been shown to be protective, often acting as opposite to AT1 receptor. The AT2 receptor inhibited cell proliferation while stimulating apoptosis in a variety of cell types, including vascular smooth muscle cells, cardiomyocytes, endothelial cells, and malignant prostate and lung cells [79,80,81,82]. Furthermore, AT2 receptor overexpression induced apoptosis independent of Ang II, through p38 MAPK, caspase 8 and caspase 3 activation, and was mediated through an extrinsic cell death-signaling pathway partially dependent on p53 [83,84]. Notably, nanoparticle-based AT2 receptor gene transfection, which markedly increased AT2 receptor expression, resulted in increased cell death of the human lung cancer cell line A549 [81]. The activation of AT2 receptor signaling has been described to produce antiproliferative, survival-promoting, as well as migratory, pro-invasive effects due to phosphatase-mediated (phosphotyrosine phosphatase PTP, protein tyrosine phosphatase 1B, PTP1B, and protein phosphatase 2A, PP2A) dephosphorylating events, through an inhibitory effect on Ras and MEK1/2 proteins of the Smad and MAPK pathways (PTP and PP2A) and by PTP1B-mediated novel Caveolin-1 (CAV1) dephosphorylation and inhibition of the CAV1/Rab5/Rac-1 pathway [69,85] (Figure 2).

#### 3.1.3. ACE2/Ang (1–7)/Mas Receptor Axis

The branch of the ACE2/Ang-(1–7)/MasR axis connected to the RAS has been associated with anti-proliferative and anti-metastatic properties. It was shown that overexpression of ACE2 inhibited the proliferation of lung cancer cells in vitro and reduced tumor growth in vivo, in a mouse lung xenograft model [86]. Furthermore, with human lung tumor xenografts evidenced that treatment with Ang-(1–7) reduces tumor volume in mice and inhibits cell proliferation through reduction on Cyclooxygenase-2 (COX-2) activity [87]. Moreover, it was demonstrated that Ang-(1–7) reduced lung cancer cell migratory and invasive abilities through the decrease of expression and activity of matrix metalloproteinases MMP-2 and MMP-9, and inactivation of the PI3K/Akt, P38 and JNK signaling pathways [88]. Noteworthy, it was recently reported that Ang-(1–7) inhibited human lung cancer cell EMT through the reduction of Cdc6 expression, dependent on the agonism of Mas receptor. An anti-metastatic mechanism, induced upon activation of ACE2/Ang-(1–7)/Mas axis has been recently described in a breast cancer model that includes repression of store-operated calcium entry (SOCE)-induced PAK1/NF-κB/Snail1 signal pathway, ultimately resulting in decreased expression of Snail1 and increased production of E-cadherin [59]. Further data from cardiomyocytes supports a nitric oxide/guanosine 3′,5′-cyclic monophosphate-dependent pathway, which modulated the activity of the transcription factor NFAT (nuclear factor of activated T-cells), preventing its translocation to the nucleus [89], and impairing the transcriptional regulation of genes involved in malignant cell proliferation, migration and metastasis [90]. However, the crosstalk between ACE2/Ang-(1–7)/Mas axis and NFAT upregulation in lung cancer has never been explored and the relationship to tumors is speculative based on studies in other tissues.

Taken together, this data provide a link to support the ACE2/Ang-(1–7)/Mas as a beneficial and targetable system for the treatment of lung cancer, reducing cell proliferation and preventing metastasis (Figure 2).

### 3.2. Hypoxia and Promotion of Tumor Angiogenesis

Hypoxia is typically present on solid tumors, such as lung cancer, and is known to enhance tumor aggressiveness, disease progression and therapy resistance. The effects of hypoxia are largely mediated by hypoxia-inducible factors (HIFs), particularly HIF-1α and HIF-2α [91]. Importantly, RAS activation at the tumor microenvironment through Ang II has been described to induce hypoxia through production of reactive oxygen species (ROS), subsequently leading to pro-inflammatory and pro-angiogenic signals (Figure 3) [11,92]. Conversely, hypoxia regulates RAS-related proteins expression in somatic tissues and cells, up-regulating Ang II, ACE, and AT1 receptor, while down-regulating ACE2 and AT2 receptor in lung carcinoma cells [93,94,95].

#### 3.2.1. ACE/Ang II/AT1 Receptor Axis and AT2 Receptor

Several studies indicated that HIF-1α may be activated by Ang II upon binding to AT1 receptor [92,96,97]. Consequently, HIF-1α not only activates vascular endothelial growth factor (VEGF) expression, but also upregulates ACE expression, which then increases endogenous Ang II formation and enhances AT1 receptor activation [98]. A significant reduction on VEGF-A mRNA and protein, in association with attenuated tumor growth, was also observed in Lewis lung tumors following candesartan administration, an ACE inhibitor. Furthermore, it was shown that treatment of Lewis lung carcinoma cells with Ang II caused a significant increase in VEGF-A mRNA and protein, which was prevented with administration of an AT1 receptor antagonist [99]. Further evidence from molecular in vitro studies demonstrated that Ang II elicited the expression of placental growth factor (PlGF) at both mRNA and protein expression in human vascular endothelial and smooth muscle cells, mediated through the activation of protein kinase C, extracellular signal-regulated kinase 1/2 (ERK1/2) and PI3-kinase [100] intracellular signaling pathways. PlGF has an important role in tumor angiogenesis [101,102], although a stimulatory effect was also reported for MMP-9-mediated NSCLC cancer cell invasion [103]. In contrast to AT1 receptor, the role of AT2 receptor in cancer angiogenesis is less clear. Ex-vivo experiments in Lewis lung tumor xenografts revealed a significant decrease in angiogenesis after inhibition of AT2 receptor or in AT2 receptor-knockout mice with impaired production of proangiogenic factors included VEGF [104].

#### 3.2.2. ACE2/Ang (1–7)/Mas Receptor Axis

ACE2 and Ang-(1–7) are known counter-regulators of ACE and Ang II activity. In vitro analysis in murine endothelioma revealed the inhibition of tumor growth after Ang-(1–7) treatment, which was related with the reduction of HIF-1A and PLGF gene expression [105]. Previous studies demonstrated that Ang-(1–7) inhibits in vitro growth of human lung cancer cells and tumor angiogenesis in vivo, through activation of the Mas Receptor [86,106]. ACE2 overexpression or ACE2-Ang-(1–7)-MasR activation may reduce angiogenesis, either through the inhibition of VEGFα production in NSCLC, or downregulation of the expression of VEGFR in nasopharyngeal carcinoma, respectively [107,108].

### 3.3. Inflammation

In lung pathologies, chronic obstructive pulmonary disease (COPD), lung fibrosis and lung inflammation are considered relevant events for cancer development [109]. Chronic inflammation is known to contribute towards the tumor microenvironment—rich in inflammatory cells, ROS, recurrent DNA damage, growth factors, and other growth-supporting stimuli. Inflammation has been postulated to play a key role in lung carcinogenesis. Polymorphisms in genes coding for inflammatory signaling molecules have also been involved in lung cancer development [110]. Engels and colleagues assessed lung cancer risk in relation to a large number of candidate polymorphisms in inflammation-related genes, namely *IL1A* and *IL1B* genes [111]. Along with a predisposing germline genetic background, environmental exposure of agents associated with elevated lung cancer risk, such as tobacco, silica or asbestos, damage the lung through the modulation of chronic inflammation. Individuals with tuberculosis, HIV infection associated with chronic lung infections, COPD and interstitial lung disease [111,112,113] seem to have an increased risk of lung cancer, reflecting the effects of underlying inflammatory disorders.

#### 3.3.1. ACE/Ang II/AT1 Receptor and AT2 Receptor

Several studies have shown the influence of RAS in inflammation [114,115,116,117]. Activation of AT1 receptor through Ang II results in potent oxidant signaling through NADPH complexes, which are involved in inflammation [11]. Within the tumor microenvironment, the major components of RAS are expressed by cancer cells as well as by stromal cells, such as macrophages and cancer-associated fibroblasts [118]. Through the upregulation of key inflammatory molecules, tumor cells invoke a chronic inflammatory state that also induces tumor-supporting myeloid cells, such as tumor-associated macrophages (TAMs) [119,120]. Notably, macrophages can exert pro-inflammatory (M1) or anti-inflammatory (M2) effects according to their polarization status. M1 macrophages are tumoricidal and highly phagocytic, promoting a Th1 response. Moreover, the tumor microenvironment can polarize M1 to M2 macrophages that have anti-inflammatory activities and promote angiogenesis, proteolysis, and tissue repair [121]. In patients with NSCLC, the M1 macrophage subset has been associated to extended survival time [122]. A recent report showed that Ang II influences macrophage behavior, fostering cancer growth in an in vivo non-small cell lung cancer model, and restraining macrophage amplification after Ang II blockade [123]. Shen and colleagues demonstrated that ACE overexpression reduced the number of blood and splenic myeloid-derived suppressor cells, in a spontaneous tumor model and in a model of chronic inflammation [124]. Furthermore, macrophages from mice with myeloid ACE overexpression were more pro-inflammatory, exhibiting enhanced tumor-killing activity compared to their wild-type mice counterparts [124].

Although there is some controversy in the field, Ang II/AT1 receptor signaling stimulates the expression of cytokines and growth factors, which enhance lung cancer-associated inflammation. Activation of AT1 receptor signaling cascade leads to ROS, prostaglandin E2 and nitric oxide production [125]. In addition, Ang II induced *Cox-2* gene expression by activating the calcineurin/NFAT signaling pathway in endometrial cells and by malignant cells (Figure 4) [126,127]. Little is known about the contribution of AT2 receptor. Its functions remain controversial, as it has been reported to either inhibit or promote inflammation in different experimental settings [128,129,130,131], albeit most studies support that AT2 receptor mainly exerts an anti-inflammatory action [132].

#### 3.3.2. ACE2/Ang (1–7)/Mas Receptor Axis

In opposite from AT1 receptor activation by Ang II, the Ang-(1–7) binding to Mas receptor triggers a down regulation of pro-oxidant pathways, preventing or attenuating the oxidative stress-induced cellular damage [114] In addition, Ang-(1–7) can also inhibit Ang II-activated inflammation through a deregulatory effect in lectin-like oxidized low-density lipoprotein receptor-1 [133]. Menon and colleagues showed that Ang-(1–7) reduces tumor volume and inhibits cell proliferation via the reduction of COX-2 activity in a human lung tumor xenograft model [87]. Moreover, compelling evidence suggests that the Mas receptor activation may counterbalance Ang-II-mediated proinflammatory effects, likely through macrophage function. A recent study reported that Mas signaling affected macrophage polarization, migration, and mediated T-cell activation, in two different chronic inflammatory animal models [134], providing the rationale that Mas may have an important role in regulating inflammatory processes, likely influencing macrophage behavior at the cancer microenvironment, namely in lung cancer. Additional experimental work in diabetes showed that administration of Ang-(1–7) limited the activation of NADPH oxidase through downregulation of molecular components of the NADPH oxidase complex [135], thus controlling ROS production and subsequent inflammation.

### 3.4. Tumor Immunological Response

Recent cancer immunotherapies, including immune-checkpoint blockade, have produced durable clinical effects in some patients with various advanced cancers, namely in NSCLC [136]. Tumor anti-programmed death-ligand 1 (PD-L1) expression is associated with increased tumor-infiltrating lymphocytes in lung cancer. Unresponsiveness to the immune-checkpoint blockade therapies may be mediated by numerous immunosuppressive mechanisms that inhibit T-cell responses and T-cell infiltration into tumors [137,138]. In the lung cancer microenvironment, the cooperation between cancer cells namely airway epithelial cells, macrophages, and other peripherally-recruited innate immune cells can determine the fate of lung tumors at different stages, of both metastatic and primary disease, including pre-neoplasic, early, and late lesions [139]. Among the hematopoietic suppressor cells of interest are M2 macrophages, splenic myeloid-derived suppressor cells, and regulatory T cells (Treg), which have been associated with a poor prognosis in many cancers, and with unfavorable clinical response to anti-PD-1/PD-L1 [137]. Furthermore, cancer-associated fibroblasts (CAFs) can manipulate the immune system, directly by inhibiting T and NK (natural killer) cell functions, promoting accumulation of suppressive cell types, and maintaining a pro-tumorigenic milieu. CAFs also induce tumor fibrosis, which represents a physical barrier to T cell infiltration [140]. Neutrophils make up a significant portion of the inflammatory cells infiltrate in the tumor, whereby they show high functional plasticity displaying both anti-tumor and pro-tumor activities [141]. Studies in early NSCLC have shown that neutrophils are pivotal to tumor cell clearance by stimulating adaptive immunity. In contrast, the role of neutrophils during later stages of primary lung cancer progression frequently appears to be pro-tumorigenic, as neutrophils represent the most abundant recruited cells in more advanced NSCLC [139]. Tumor-derived signals induce a pro-tumor phenotype in neutrophils, which supports cancer cell invasion and metastasis (N2 neutrophils). N2 polarized neutrophils promote the proliferation, migration, and invasion of tumor cells, stimulate angiogenesis, as well as mediate immunosuppression [141]. Dendritic cells are crucial for the activation of antigen-specific CD4 T lymphocytes, a pivotal step in the initiation of innate and adaptive immune responses, which are essential for lung tumor cell clearance [142]. Previous studies demonstrated that lung tumors dynamically exclude functional DCs from the tumor region to support malignant progression. Furthermore, clinical trials have proven that dendritic cell function is reduced in lung cancer patients [142]. At the tumor microenvironment, the major components of RAS are also expressed both by resident and infiltrating cells, such as endothelial cells, fibroblasts, monocytes, macrophages, neutrophils, dendritic and T cells [11]. RAS signaling in these cells can facilitate or hinder growth and dissemination and has been shown to affect cell proliferation, migration, invasion, metastasis, apoptosis, angiogenesis, cancer-associated inflammation, immunomodulation, and tumor fibrosis/desmoplasia [142].

#### Targeting RAS to Improve Anti-Tumor Immunity

A recent study using 4T1 and CT26 syngeneic mouse tumor models, showed that Ang II generates infiltration of fibroblasts and macrophages, contributing to the reduction of CD8^+^ T cell infiltration, resulting in an immunosuppressive environment [143]. Importantly, Nakamura and colleagues, demonstrated, in a tumor bearing mouse model of colorectal carcinoma, that the antagonism of AT1 receptor decreased the expression of immunosuppressive factors, such as chemokine ligand 12 and nitric oxide synthase 2 in CAFs, and that the combination of antagonists of AT1-receptor and PD-L1 antibodies resulted in significant augmentation of anti-tumor effects in a CD8^+^ T cell-dependent way [118]. Furthermore, RAS-activated CAFs secrete collagen and other ECM components which produce fibrosis and consequently tumor desmoplasia [144]. Notably, tumor desmoplasia can be either a physical barrier to immune cell infiltration or provide the substratum to interstitial cell migration [144]. It was demonstrated that the antagonism of AT1 receptor-inhibited collagen I production by carcinoma-associated fibroblasts isolated from breast cancer biopsies [145]. Additionally, this led to a dose-dependent reduction of stromal collagen in desmoplastic models of human breast, pancreatic, and skin tumors in mice [145]. Contrary to the Ang II effect in tumor immunity response, interesting findings also showed that Ang-(1–7) targets the tumor microenvironment to inhibit CAF growth and tumor fibrosis in orthotopic breast tumors [146]. TGF-β suppresses the differentiation and function of T helper, CD8^+^ cells, Natural Killer cells, and tumor-associated neutrophils, tumor associated macrophages and myeloid-derived suppressor cells [147]. Tumor supporting cytokines are released from tumor and stromal cells upon AT1 receptor activation via Ang II including, TGF-β and Interleukins (IL-1α, IL-1β, IL-6, IL-8) [140]. Immunomodulatory cytokines may up-regulate immunosuppressive pathways, i.e., COX-2 via prostaglandin E2 synthesis, and impair dendritic cell function by reducing their migration [148]. Ang II/AT1R signaling induces reactive oxygen species generation and related proteins such as inducible nitric oxide synthase in tumor and stroma cells. Exposure to ROS in the tumor microenvironment can impair T cell function while enhancing T regs and tumor associated macrophages [140]. Treatment with the candesartan (AT1 receptor blocker) diminishes ROS generation [144]. Ang II/AT1R signaling is also important for myeloid differentiation and functional maturation. Cultured bone marrow from ACE 10/10 mice, a mouse line overexpressing ACE in monocytic cells, demonstrates increased myeloid maturation and reduced splenic myeloid-derived suppressor cells production. In addition, macrophages from these mice have an enhanced proinflammatory phenotype and antitumor activity compared to those from wild-type mice [124]. Similarly, tumor-bearing ACE 10/10 mice showed enhanced immune responses, which ultimately resulted in a reduced tumor growth. ACE inhibitors reversed the beneficial effects on tumor growth, but AT1 receptor blockade did not, suggesting that the effects of ACE overexpression were not dependent of Ang II/AT1 receptor signaling [149]. Hence, these reports suggest that targeting RAS could alleviate immunosuppression and enhance the outcome of immunotherapy. Nevertheless, it is necessary, through experimental and clinical research, to highlight the role of RAS in lung cancer and its influence on immunotherapy response.

## 4. Clinical Studies

Pre-clinical studies demonstrated an association between RAS activation and increased invasion, angiogenesis, inflammation, and modulation of immune response at the lung tumor microenvironment. The putative impact on cancer hallmarks, through AT1 receptor blockade or activation of the ACE2/Ang-(1–7)/Mas receptor axis, has been linked to reduced tumor growth and vascularization in lung cancer, thus fostering the need for clinical studies.

### 4.1. Clinical Trials

Despite basic and pre-clinical evidence concerning the impact of RAS in cancer hallmarks, the information available regarding clinical trials remains scarce, particularly in lung cancer. The primary objective of a Phase II Randomized Trial (NCT00077064, Appendix A) was to test the ability of captopril to alter the incidence of pulmonary damage at 12 months after the completion of radiation in patients with non-small cell lung cancer or limited-stage small cell lung cancer [150]. Due to the unmet accrual/randomization goals, it was not possible to adequately test the hypothesis that captopril ameliorates radiotherapy-induced pulmonary toxicity. Another double-blind placebo-controlled randomized trial (NCT01880528, Appendix A) enrolled 23 patients with lung cancer to study the putative protective effect of lisinopril in pneumonitis induced by radiotherapy [151]. The results of this clinical trial suggest that there was a clinical signal for safety and possibly beneficial in concurrently administering lisinopril during thoracic radiotherapy to mitigate or prevent radiation-induced pulmonary distress [151]. However, larger-scale randomized phase 3 trials are needed in the future to confirm these results. Notably, a multicenter clinical trial showed that losartan stabilized lung function in patients with idiopathic pulmonary fibrosis over 12 months (NCT00879879, Appendix A), reinforcing the importance of AT1 receptor blockers in the impairment of fibrosis progression [152]. Moreover, treatment with AT1 receptor blockers might potentiate the response to immunotherapy in lung cancer patients due to a hypothetic impairment of the fibrotic immunosuppressive microenvironment. Nevertheless, there is no study still to prove this hypothesis.

A small Phase I study (NCT00471562, Appendix A) of Ang-(1–7) as a first-in-class anti-angiogenic drug targeting Mas receptor, was undertaken in 18 patients with advanced solid tumors refractory to standard therapy with only one patient with lung cancer included [153]. Dose-limiting toxicities encountered at the 700 μg/kg dose included stroke (grade 4) and reversible cranial neuropathy (grade 3). Other toxicities were generally mild. Clinical benefit, defined as disease stabilization for more than three months, was observed in two of the three patients with metastatic sarcoma and it was associated with reduction of PlGF plasma levels [153]. A Phase II study failed to confirm PlGF as biomarker of response to Ang-(1–7) administration [154]. However, two patients with vascular sarcomas demonstrated prolonged disease stabilization of 10 months (hemangiopericytoma) and 19 months (epithelioid hemangioendothelioma) under Ang-(1–7) treatment, thereby requiring further investigation [154].

### 4.2. Observational Studies

A retrospective cohort study that included 228 patients with advanced NSCLC and 73 with early stage disease revealed that those under therapeutics with ACE inhibitors (iACE) or AT1 receptor blockers, in combination with standard chemotherapy or tyrosine kinase inhibitors, had a positive effect on progression-free survival and overall survival, regardless of lung cancer stage [65]. A retrospective study with 117 metastatic NSCLC patients demonstrated that the intake of AT1 receptor blockers, during erlotinib treatment, may prolong overall survival [155]. Another study including 287 advanced NSCLC patients, demonstrated a survival advantage for patients under ACE inhibitors or AT1 receptor blockers [156]. Notably, the reduced risk for cancer progression in patients submitted to ACE inhibitors or AT1 receptor blockers has not been seen with other antihypertensive medications, suggesting the existence of an adjuvant effect for angiotensin blockers [156]. A recent systematic review and meta-analysis showed that AT1 receptor blockers are significantly associated with reduced lung cancer risk [157]. Conversely, a study on a large population-based cohort that used ACE inhibitors suggested an increased risk for developing lung cancer, particularly among patients undergoing treatment for more than five years. Nevertheless, the risk for lung cancer was not confirmed in patients under AT1 receptor blockers [158]. These findings might rely on ACE inhibitors-associated accumulation of bradykinin in the lung, rather than a decrease in Ang II signaling, especially when higher dosages of ACE inhibitors are taken into consideration [159]. When analyzed together, these results from clinical and pre-clinical studies suggest that AT1 receptor blockers, in contrast to ACE inhibitors, might be putative therapeutic tools to impair lung cancer progression. Further larger prospective studies with well-designed clinical trials are required to confirm whether targeting RAS improves lung cancer treatment.

## 5. Future Perspectives

A significant body of knowledge supports a role for the RAS as modulator of the tumor microenvironment through direct and indirect influences on tumors and surrounding cells, including immunological mediators. In vitro and in vivo studies demonstrate that the activation of the ACE/Ang II/AT1 receptor axis promotes cancer cell proliferation, invasion, angiogenesis and reduced immunosurveillance, albeit opposing effects were found when the ACE2/Ang-(1–7)/Mas receptor was activated. Nevertheless, those pathways remain largely unexplored at translational and clinical levels, prompting the need for further research. The major challenge in clinical context is the complex nature of RAS signaling, which seems to be variable when considering several factors, making the response to RAS therapeutics, either individually or in combination with other drugs, difficult to predict [11]. Furthermore, dysregulation of RAS can occur due to germline [160] or somatic mutations [161], or due to hypoxia within the tumor microenvironment [95]. Putative functional genetic variants coding for proteins of RAS might have predictive and/or prognostic value in lung cancer, opening novel opportunities to guide clinical reasoning and therapeutics in lung cancer patients.

Moreover, to confirm whether targeting the RAS has the potential to reprogram cancer cell immunogenicity and the immunosuppressive lung tumor microenvironment, preclinical research that combines immune checkpoint inhibitors with RAS agonists/antagonists is needed. Finally, the intra-tumor expression of the components of the RAS may also add potential to predict the response to RAS therapeutics.

## 6. Conclusions

Malignant transformation and lung tumor progression is associated with at least six acquired, functional capabilities: sustained angiogenesis, evasion of apoptosis, self-sufficiency in growth signals, insensitivity to anti-growth signals, tissue invasion and metastasis, and limitless replication potential [162]. The RAS regulates all these capabilities, although most prominent are its effects on angiogenesis, invasion, pro-survival signaling and proliferation. Inevitably, many of these processes are interdependent and cooperative [162]. Preclinical studies have provided compelling evidence that the Ang II/AT1R axis regulates almost all hallmarks of cancer. It is also clear that the Ang II/AT1R signaling contributes to the immuno-suppressive tumor microenvironment in multiple ways. The immuno-suppressive milieu is a major barrier for immunotherapy and may explain why immune checkpoint inhibitors failed in NSCLC patients [140]. The high expression of the gene encoding the AT1 receptor in lung tumors suggests that Ang II acting upon the AT1 receptor plays an important role in cell growth, invasion, fibrosis and angiogenesis. The low level of expression of the genes encoding the components of the ACE2/Ang 1–7/MasR axis in lung tumors promotes the tumorigenic effects of Ang II, while reducing the stimulation of the Mas receptor by Ang 1–7, that has contradictory Ang II/ATR1 mediated effects [35]. Multiple clinical studies have also revealed that RAS inhibitors may have beneficial effects in a broad range of malignancies. The gain in survival is tumor type-and-stage-dependent and ranged from 3 months (advanced NSCLC) to more than 25 months (metastatic colorectal cancer) in retrospective studies [140]. Finally, taking the large amount of preclinical and clinical data suggesting a beneficial effect of RAS inhibitors in different cancer types, including lung cancer, into account, we indicate that RAS inhibitors have a great potential to become an adjunct within the oncological armamentarium.

## Figures and Tables

**Figure 1 cancers-12-01457-f001:**
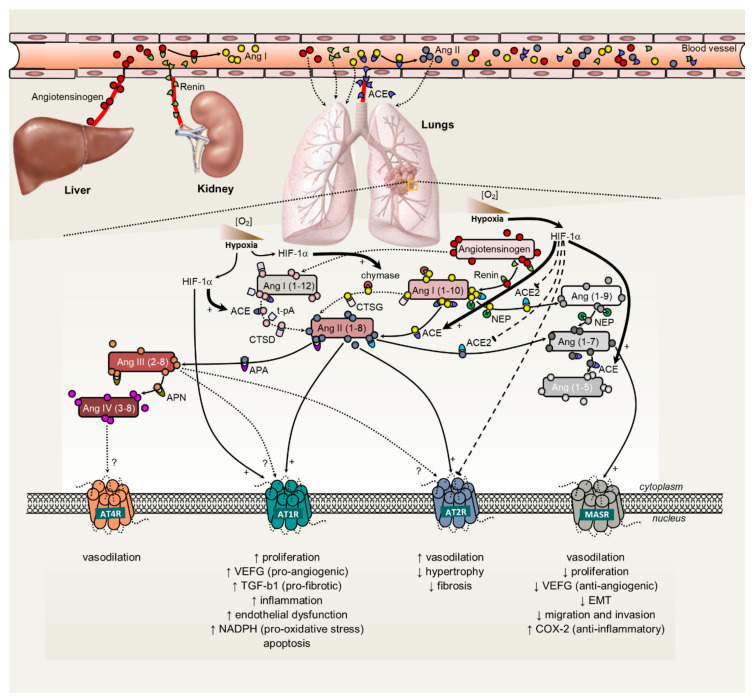
Schematic representation of the renin-angiotensin system, its interaction with tumor hypoxia and angiogenesis and subsequent functional impact on tumor phenotype. Full line arrows represent positive effects, whereas dashed lines with blunt ends represent inhibitory effects. Angiotensinogen, a precursor of angiotensin peptides, is the only known naturally occurring renin substrate. Ang I is processed by ACE or ACE2 to produce Ang II and Ang-(1–7), respectively. Alternatively, Ang I can be converted to Ang II through chymase. Ang-(1–7) can be generated from Ang I, via Ang-(1–9), a pathway that utilizes both ACE2 and ACE, or by neprilysin that hydrolyzes Ang I directly to Ang-(1–7). Ang II acts in cells through two classes of seven trans-membrane G protein-coupled receptors: the angiotensin type I (AT1) and angiotensin type II (AT2) receptors. Upon binding of Ang II to ATR1, a downstream intracellular cascade is induced that results in cell proliferation, fibrosis and hypoxia by generating reactive oxygen species and subsequently pro-inflammatory and pro-angiogenic signals. The AT1 receptor is overexpressed in neoplastic tissues, suggesting its involvement in carcinogenesis. The role of AT2 receptors in cancer remains controversial, although including pro-apoptotic and anti-proliferative effects. Ang-(1–7) binds to MasR and counteracts the resulting effects of Ang II/AT1 receptor activation, preventing tissue remodeling, improving vascular and cardiac function, and promoting the downregulation of cell proliferation, migration and metastasis. Hypoxia up-regulates Ang II, ACE, and AT1 receptor, while downregulating ACE2 and AT2 receptor. Local angiotensin II predominantly exists in the hypoxic regions of tumors, while these tumor cells produce Ang II by a chymase-dependent mechanism. Subsequently, the hypoxic tumor microenvironment induces the upregulation of the ACE/Ang II/AT1R axis, which is associated with the hallmarks of cancer, while it downregulates the ACE2/Ang-(1–7)/Mas receptor axis. ACE, angiotensin converting enzyme; ACE2, angiotensin converting enzyme 2; AMNA, aminopeptidase A; AMNN, aminopeptidase N; Ang, angiotensin; AT1R, angiotensin receptor 1; AT2R, angiotensin receptor 2; AT4R, angiotensin receptor 4; CAGE, chymostatin-sensitive angiotensin II-generating enzyme; COX-2, cyclooxygenase 2; EMT, epithelial-to-mesenchymal transition; HIF1α, hypoxia-inducible factor 1-alpha; MASR, Mas receptor; NADPH, hydro-nicotinamide adenine dinucleotide phosphate; TGFβ1, transforming growth factor beta 1; t-PA, tissue-plasminogen activator; VEGF, vascular endothelial growth factor.

**Figure 2 cancers-12-01457-f002:**
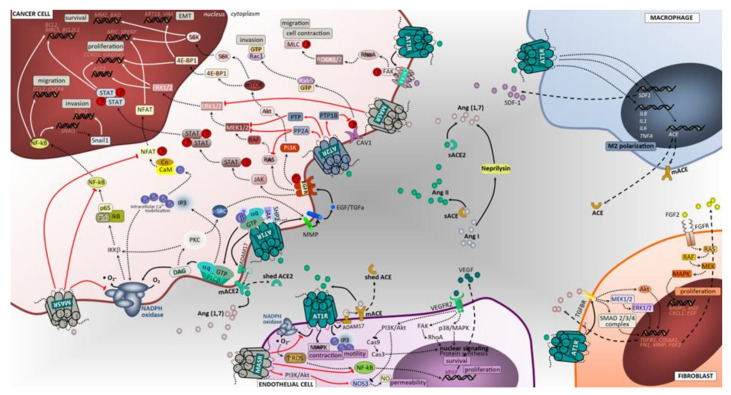
Angiotensin-associated pathways associated with cell proliferation, invasion, and migration in lung tumor microenvironment. Membrane-bound and soluble ACE and ACE2 catalyze the production of angiotensin II or angiotensin (1,7), ligands that exert regulatory functions in tumor microenvironment cells. The effects of activating Ang II/AT1R, Ang II/AT2R and Ang (1,7)/MasR axes’ signaling has been mostly studied in tumor cells, including lung cancer cells. Overall, in contrast to the Ang II/AT1R that mediates several pathological events associated with activated RAS, the Ang (1,7)/MasR and Ang II/AT2R pathways are thought to antagonize many of the cellular actions of the Ang II-AT1R axis. In cancer cells, upon binding of Ang II to AT1R, a pleiotropic downstream signaling cascade is triggered, ultimately causing, either directly or indirectly, upregulation of cell proliferation, survival, motility, migration, invasion and EMT. Activated AT1R subunits stimulate PLCβ, that hydrolyses membrane lipids, activates PKC and mobilization of intracellular Ca^2+^, while free Gβ and Gγ subunits bind and gate ion channels. Activated Gαq/11 units also activate the JAK-SHP2/STAT pathway and receptor tyrosine kinase (RTK) transactivation. Activation of RTK, depicted in the figure as EGFR, occurs through second messenger’s stimulation of ADAM family and MMPs to cleave its ligands (in the figure EGF and TGFα) that bind and activate RTK. Alternatively, the AT1R-mediated activation of MMPs can follow a PLCβ/DAG/PKC/c-SRC signaling mechanism to elicit increased ligands for RTK. Subsequent signaling upon activation of EGFR is represented in the figure using dashed arrows. Other intracellular cascades mediated by Ang II/AT1R include the activation of CXCR4/SDF-1 signaling through FAK/RhoA/ROCK1-2/MLC increasing cell contraction, migratory potential and tumor invasion. The Ang (1,7)/MasR activation inhibits NFAT transcriptional regulation that reduces proliferation. Notably, this pathway blocks the NF-kB molecule formation thereby impacting EMT (including Snail1-mediated), invasion and survival. In addition, the inhibitory effect over ERK1/2 and NAPH oxidase signaling pathways significantly impact cell proliferation. The Ang II/AT2R pathway signals are mediated through protein phosphatases PTP1B, PTP and PP2A. The inhibition of CAV-1 phosphorylation stops the Rab5/Rac1/GTP migratory potential of malignant cells, thus suppressing invasion. Furthermore, AT2R-associated increase in PTP and PP2A exerts blocking effects in RTK-mediated signals of the RAS/RAF/MEK1-2/ERK1-2 pathway at the level of MERK1/2 and RAS molecules, reducing cell proliferation. Macrophages, endothelial cells and fibroblasts are important components of the tumor microenvironment and capable of generating and expressing RAS components. These cells, beyond their functional ligands and receptors that are altered in tumor microenvironment, and reflect the crosstalk between all cell constituents, also use the RAS signaling pathway (mostly AT1R, but also MasR in endothelial cells) to yield functional characteristics that ultimately may favor the cell itself and enhance tumor growth. BIRC5, surviving gene; BCL2, B-cell leukemia/lymphoma 2; BAD, BCL2 associated agonist of cell death; GADD45, Growth Arrest and DNA Damage 45; SREBP, Sterol regulatory element binding proteins; ROCK1/2, Rho-associated protein kinase 1/2; Rab5, Ras-related protein Rab-5; S6K, ribosomal S6 kinase; PTP, protein tyrosine phosphatase; PP2A, Protein phosphatase 2; 4E-BP1, Eukaryotic translation initiation factor 4E-binding protein 1; IKKB, Inhibitor of nuclear factor kappa-B kinase subunit beta; c-SRC, cellular Proto-oncogene tyrosine-protein kinase; SHP2, Src homology region 2 domain-containing phosphatase-2; PTP1B, Protein tyrosine phosphatase 1B; MLC, myosin light chain; CAV1, caveolin 1; IL1, interleukin 1; TNFA, tumor necrosis factor; Snail1, Zinc finger protein SNAI1; *SNAI1*, Snail Family Transcriptional Repressor 1; JAK, janus kinase; STAT, signal transduction and activator of transcription; GTP, guanosine triphosphate; EGF, epidermal growth factor; EGFR, epidermal growth factor receptor; CXCL1, The chemokine (C-X-C motif) ligand 1; IL8, interleukin 8; NF-kB, Nuclear Factor-kappa B; SDF1, stromal cell-derived factor 1; IL6, interleukin 6; HGF, hepatocyte growth factor; RAS, oncogene protein p21; FGF, fibroblast growth factor; FGFR, fibroblast growth factor receptor; Cas9, caspase 9; Cas3, caspase 3; ERK1/2, extracellular signal-regulated kinase; PI3K/Akt, phosphoinositide 3-kinase/protein kinase B; MEK1/2, mitogen activated protein kinase kinase; HIF1α, hypoxia inducible factor 1 alpha; NO, nitric oxide; RhoA, Ras homolog family member A; MAPK, mitogen activated protein kinase; mTOR, mammalian target of rapamycin; IP3, inositol trisphosphate; DAG, diacylglycerol; M2, macrophage polarized towards M2; ADAM17, Desintegrin and metalloproteinase domain-containing protein 17; p38/MAPK, protein 38/mitogen activated protein kinase; AT1R, angiotensin receptor 1; AT2R, angiotensin receptor 2; MasR, G-protein coupled Mas receptor; VEGF, vascular endothelial growth factor; NADPH, reduced form of nicotinamide adenine dinucleotide phosphate; mACE, membrane angiotensin converting enzyme; sACE, soluble angiotensin converting enzyme; mACE2, membrane angiotensin converting enzyme 2; sACE2, soluble angiotensin converting enzyme 2; TGFBR, transforming growth factor beta receptor; VEGFR2, vascular endothelial growth factor receptor 2; TGFα, transforming growth factor alpha; Ang (1,7), angiotensin 1,7; PLCβ, phospholipase C beta; FAK, Focal adhesion kinase; NFAT, nuclear factor of activated T cells.

**Figure 3 cancers-12-01457-f003:**
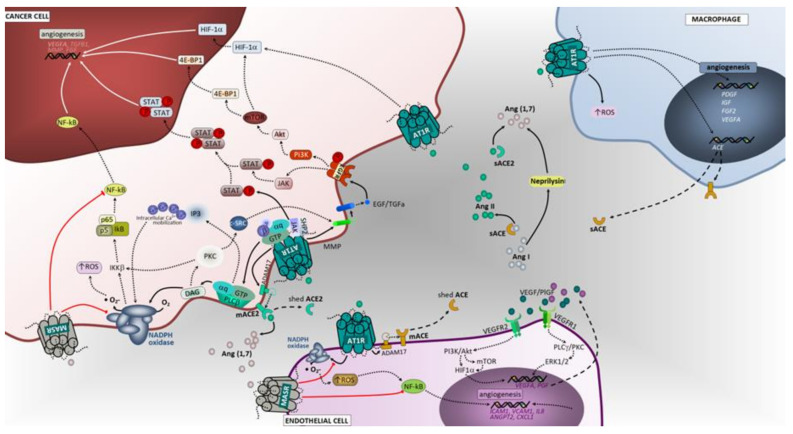
Angiotensin-associated pathways involved on hypoxia and angiogenesis in lung tumor microenvironment. In cancer cells, upon binding of Ang II to AT1R, a pleiotropic downstream signaling cascade is triggered, ultimately causing, either directly or indirectly, upregulation of angiogenesis and metastasis. Activated AT1R subunits trigger potent oxidant signaling through NADPH complex, which is involved in angiogenesis. Activated Gαq/11 units also activate the JAK-SHP2/STAT pathway and receptor tyrosine kinase (RTK) transactivation. Subsequent signaling upon activation of EGFR is herein represented. Macrophages, endothelial cells and fibroblasts are important components of the tumor microenvironment and capable of generating and expressing RAS components. These cells, beyond their functional ligands and receptors that are altered in tumor microenvironment, and reflect the crosstalk between all cell constituents, also use the RAS signaling pathway (mostly AT1, but also MasR in endothelial cells) to yield functional characteristics that ultimately may favor the cell itself and enhancing tumor angiogenesis and metastasis. 4E-BP1, Eukaryotic translation initiation factor 4E-binding protein 1; IKKB, Inhibitor of nuclear factor kappa-B kinase subunit beta; c-SRC, cellular Proto-oncogene tyrosine-protein kinase; SHP2, Src homology region 2 domain-containing phosphatase-2; IL1, interleukin 1; JAK, janus kinase; STAT, signal transduction and activator of transcription; Ang I, angiotensin I; Ang II, angiotensin II; EGF, epidermal growth factor; EGFR, epidermal growth factor receptor; ANGPT2, angiopoietin 2; ICAM1, Intercellular Adhesion Molecule 1; VCAM1, vascular cell adhesion molecule 1; NF-kB, Nuclear Factor-kappa B; ROS, reactive oxygen species; ERK1/2, extracellular signal-regulated kinase; PLCγ/PKC, phospholipase C gamma/protein kinase C; PI3K/Akt, phosphoinositide 3-kinase/protein kinase B; MEK1/2, mitogen activated protein kinase kinase; HIF1α, hypoxia inducible factor 1 alpha; NO, nitric oxide; MAPK, mitogen activated protein kinase; mTOR, mammalian target of rapamycin; IP3, inositol trisphosphate; DAG, diacylglycerol; M2, macrophage polarized towards M2; ADAM17, Disintegrin and metalloproteinase domain-containing protein 17; AT1R, angiotensin receptor 1; AT2R, angiotensin receptor 2; MasR, G-protein coupled Mas receptor; VEGF, vascular endothelial growth factor; NADPH, reduced form of nicotinamide adenine dinucleotide phosphate; mACE, membrane angiotensin converting enzyme; sACE, soluble angiotensin converting enzyme; mACE2, membrane angiotensin converting enzyme 2; sACE2, soluble angiotensin converting enzyme 2; VEGFR1, vascular endothelial growth factor receptor 1; VEGFR2, vascular endothelial growth factor receptor 2 Ang (1,7), angiotensin 1,7; PLCβ, phospholipase C beta.

**Figure 4 cancers-12-01457-f004:**
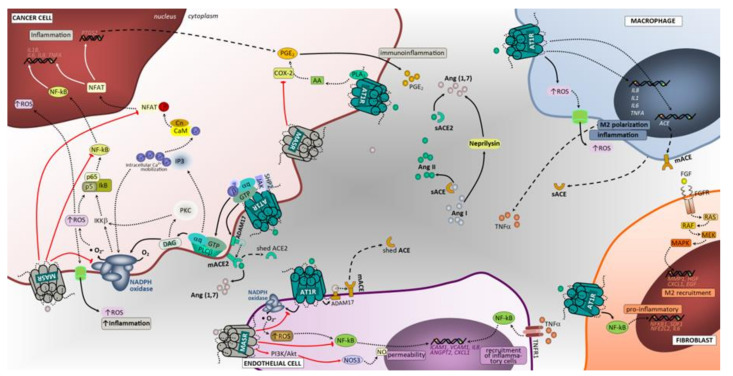
Angiotensin-associated immunoinflammatory pathways in lung tumor microenvironment. Extracellular proteins of the RAS, ligands (Ang I, Ang II and Ang 1,7) and enzymes (ACE, ACE2 and neprilysin), are represented with distinct colors in the tumor microenvironment. The expression and cleavage of ACE and ACE2 are represented in tumor and endothelial cells, showing the AT1R signaling-associated activation of ADAM17 that cleaves the ectodomain of mACE and mACE2 shedding the enzymes into sACE and sACE2. The Ang II/AT1R-mediated direct intracellular signaling is represented with full black arrows as a first step after AT1R triggering in the cascade. The subsequent steps of signaling cascades are depicted, up to moving towards intranuclear space, using dashed black arrows. Intranuclear signaling from these pathways mainly results in transcriptional regulatory effects and is represented by full white arrows. The Ang II/AT2R and Ang (1,7)/MasR-mediated counter-regulatory mechanisms are depicted using black arrows and when adequate the suppressive step is represented by blunt-ended red arrows. In cancer cells, upon binding of Ang II to AT1R, a pleiotropic downstream signaling cascade is triggered, ultimately causing, either directly or indirectly inflammation immune cells recruitment. Activated AT1R triggers potent oxidant signaling through NADPH complex, which is involved in inflammation and angiogenesis. Other intracellular cascades mediated by Ang II/AT1R include the activation of PLA2/AA/COX-2/PGE2 signaling. The Ang (1,7)/MasR activation elicits downstream signaling through PI3K/Akt/NOS3 or NOS1 to produce NO that inhibits NFAT transcriptional regulation that reduces inflammation. Notably, this pathway blocks the NF-kB molecule formation thereby impacting inflammation. In addition, the inhibitory effect over COX-2 significantly impact inflammation. Macrophages, endothelial cells and fibroblasts are important components of the tumor microenvironment and capable of generating and expressing RAS components. These cells, beyond their functional ligands and receptors that are altered in tumor microenvironment, and reflect the crosstalk between all cell constituents, also use the RAS signaling pathway (mostly AT1, but also MasR in endothelial cells) to yield functional characteristics that ultimately may favor the cell itself and enhance tumor growth and aggressiveness. PLA2, phospholipase A2; PTP, protein tyrosine phosphatase; PP2A, Protein phosphatase 2; IKKB, Inhibitor of nuclear factor kappa-B kinase subunit beta; CARMA3, CARD recruited membrane associated protein 3; MALT1, Mucosa-associated lymphoid tissue lymphoma translocation protein 1; BCL-10, B-cell lymphoma/leukemia 10; c-SRC, cellular Proto-oncogene tyrosine-protein kinase; SHP2, Src homology region 2 domain-containing phosphatase-2; AA, arachidonic acid; COX-2, cyclooxygenase 2; PGE2, prostaglandin E2; IL1, interleukin 1; TNFA, tumor necrosis factor; Ang I, angiotensin I; Ang II, angiotensin II; GTP, guanosine triphosphate; EGF, epidermal growth factor; EGFR, epidermal growth factor receptor; ANGPT2, angiopoietin 2; ICAM1, Intercellular Adhesion Molecule 1; VCAM1, *vascular cell adhesion molecule 1*; CXCL1, The chemokine (C-X-C motif) ligand 1; IL8, interleukin 8; NF-kB, Nuclear Factor-kappa B; NFKB1, Nuclear Factor-kappa B Subunit 1; IL6, interleukin 6; NFE2L2, Nuclear Factor, Erythroid 2 Like 2; HGF, hepatocyte growth factor; RAS, oncogene protein p21; ROS, reactive oxygen species; NOS3, nitric oxide synthase 3 endothelial; NOS1, nitric oxide synthase 1 neuronal; Cas9, caspase 9; Cas3, caspase 3; ERK1/2, extracellular signal-regulated kinase; PLCγ/PKC, phospholipase C gamma/protein kinase C; PI3K/Akt, phosphoinositide 3-kinase/protein kinase B; TNFR1, tumor necrosis factor receptor 1; c-Raf, kinase Raf-1; MEK1/2, mitogen activated protein kinase kinase; HIF1α, hypoxia inducible factor 1 alpha; NO, nitric oxide; RhoA, Ras homolog family member A; IP3, inositol trisphosphate; DAG, diacylglycerol; M2, macrophage polarized towards M2; ADAM17, Disintegrin and metalloproteinase domain-containing protein 17; p38/MAPK, protein 38/mitogen activated protein kinase; AT1R, angiotensin receptor 1; AT2R, angiotensin receptor 2; MASR, G-protein coupled Mas receptor; VEGFA, vascular endothelial growth factor A; NADPH, reduced form of nicotinamide adenine dinucleotide phosphate; mACE, membrane angiotensin converting enzyme; sACE, soluble angiotensin converting enzyme; mACE2, membrane angiotensin converting enzyme 2; sACE2, soluble angiotensin converting enzyme 2; PDGFRβ, platelet derived growth factor receptor beta; VEGFR1, vascular endothelial growth factor receptor 1; VEGFR2, vascular endothelial growth factor receptor 2; TGFα, transforming growth factor alpha; Ang (1,7), angiotensin 1,7; PLCβ, phospholipase C beta; FAK, Focal adhesion kinase; NFAT, nuclear factor of activated T cells; PDGF, platelet derived growth factor.

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
