# Peer review of "Renin-Angiotensin System in Lung Tumor and Microenvironment Interactions"

_cancers, 2020, doi:10.3390/cancers12061457_

Round 1

Reviewer 1 Report

The title of the review sounds appealing, with a topic being as relevant as complex, involving multiple physiological disciplines.

The structure of the review includes an extensive introduction on the "renin-angiotensin axis" and its role in the lung. Of note, the naming of the "renin-angiotensin axis" should be changed to "renin-angiotesnin-system (RAS)" instead, being the widely used and accepted terminology in literature.

Rather than being localized to certain tissues, the RAS is a systemic humoral peptide hormone cascade, which is locally modulated at tissue sites by local expression of angiotensin processing enzymes. Moreover receptor expression modulate the activity of angiotensins in their target organs. All angiotensins in the body appear to be derived from angiotensinogen prodiced in the liver, which becomes evident considering even kidney Ang II levels to disappear in liver specific knockout animals for the pro-hormone angiotensinogen (https://www.ncbi.nlm.nih.gov/pmc/articles/PMC3380650/).

Considering the local RAS in a tumor it would be essential to put more emphasis on the role of the circulating RAS (Angiotensinogen, Renin, ACE,...) in providing the substrates for local modulation of peptide hormone levels. The role of the kidney and the liver should be discussed, considering lung cancer affecting the entire organism especially at later disease stages. These aspects might be included into Figure S1. After all, the static view of Ang II being produced primary by the lung leading to circulating levels, has never been clearly proven in experimental setups. Although it is true that ACE is highly expressed by the lung, however, rather than locally producing Ang II from Ang I, shedding results in high levels of circulating ACE, which leads to sustained Ang II formation in the plasma compartment throughout the body. The crosstalk between the systemic and local RAS components are essential for the understanding of the tumor microenvironment related to the RAS, and should be elaborated in much more detail, probably including a clear figure illustration.

The authors further discuss individual molecules of the RAS in the context of cancer hallmarks

  • Proliferation, Invasion, Migration
  • Hypoxia and Angiogenesis
  • Inflammation
  • Tumor Immunological Response

While comprehensively reviewing published molecular finding in these fields, the authors failed to provide an understandable overall picture of these processes, that might help the reader to gain insight. These hallmarks and also the the role of the RAS within these, are critically dependent on the state of disease.

Finally, the figures presented by the authors are much too crowded. Trying to combine all aspects discussed previously, Figure S2 shows a very complex network of individual molecular players without transporting a clear message. 

It appears reasonable, considering the complexity of each of the hallmarks discussed, to provide separate Figures for these, which would definitely help not to loose the reader. 

Author Response

Title: Detailed Response to the Editor and to Reviewer 1 for the Manuscript ID: cancers-791140

Firstly, we would like to thank you for your constructive comments in this round of review. Your comments provided valuable insights to refine its contents and analysis. In this document, we try to address the issues raised as best as possible.

Point 1: The structure of the review includes an extensive introduction on the "renin-angiotensin axis" and its role in the lung. Of note, the naming of the "renin-angiotensin axis" should be changed to "renin-angiotesnin-system (RAS)" instead, being the widely used and accepted terminology in literature.

Answer: The authors formulated the title: “Renin–angiotensin system in lung tumour and its microenvironment interactions”.

Point 2: Rather than being localized to certain tissues, the RAS is a systemic humoral peptide hormone cascade, which is locally modulated at tissue sites by local expression of angiotensin processing enzymes. Moreover receptor expression modulate the activity of angiotensins in their target organs. All angiotensins in the body appear to be derived from angiotensinogen prodiced in the liver, which becomes evident considering even kidney Ang II levels to disappear in liver specific knockout animals for the pro-hormone angiotensinogen (https://www.ncbi.nlm.nih.gov/pmc/articles/PMC3380650/).

Answer: It is well known that the in vivo generation of angiotensin is dependent on both plasma renin and angiotensinogen concentrations. The authors improved the description of the importance of angiotensins derived from angiotensinogen and that is produced in the liver.

Line 51: “In fact, the liver is the primary source of circulating Angiotensinogen (AGT), a minor but significant fraction of which is filtered through the glomerulus and reabsorbed by proximal tubule cells. It was shown that liver-specific angiotensinogen knockout mice had nearly abolished plasma and renal angiotensinogen protein and renal tissue angiotensin II (https://www.ncbi.nlm.nih.gov/pmc/articles/PMC3380650/)”.

Point 3: Considering the local RAS in a tumor it would be essential to put more emphasis on the role of the circulating RAS (Angiotensinogen, Renin, ACE,...) in providing the substrates for local modulation of peptide hormone levels. The role of the kidney and the liver should be discussed, considering lung cancer affecting the entire organism especially at later disease stages. These aspects might be included into Figure S1. After all, the static view of Ang II being produced primary by the lung leading to circulating levels, has never been clearly proven in experimental setups. Although it is true that ACE is highly expressed by the lung, however, rather than locally producing Ang II from Ang I, shedding results in high levels of circulating ACE, which leads to sustained Ang II formation in the plasma compartment throughout the body. The crosstalk between the systemic and local RAS components are essential for the understanding of the tumor microenvironment related to the RAS, and should be elaborated in much more detail, probably including a clear figure illustration.

Answer: In fact the circulating RAS might be important in providing the substrates for local modulation of angiotensin peptides. Genetic and epidemiological evidence provides support that germline mutations in RAS components contribute to the risk of developing certain malignancies. Those functional germline mutations influence the expression levels of circulating components of the RAS, although more studies are necessary to confirm these results. However, the circulating RAS is mainly known for its pivotal role in maintaining cardiovascular homeostasis and fluid and electrolyte balance. Local RAS is expressed in many tissues and mainly acts at the cellular level, where it mediates cell proliferation, growth, and metabolism. This local RAS works synergistically and independently of the systemic RA. The dysregulation of RAS occurs mainly in the tumour microenvironment and depends on the type of cancer. For example, a large-scale study of estrogen receptor-positive breast cancer tumours revealed an increase in AGTR1 mRNA expression, which is the opposite of what we observed in the lung cancer study from which this data is derived [PMID 28791183]. Equally contradictory, AGTR1 as well as ACE mRNA expression is upregulated in hormone-independent prostate cancer [PMID 28791183].

The authors explored the role of the circulating RAS deregulation in lung diseases including lung cancer: Line 114: “The circulating RAS might also be important in providing the substrates for local modulation of angiotensin peptides with influence in lung diseases. Genetic and epidemiological evidence provides support that germline mutations in RAS components contribute to the risk of developing idiopathic pulmonary fibrosis, acute lung injury and certain malignancies. For example, the human gene encoding angiotensinogen (AGT) is located on chromosome 1q42.3 and a large number of single-nucleotide polymorphisms (SNPs) have been described. Among them, the SNPG-6A nucleotide substitution at position -6 upstream from the initial transcription start has been studied particularly. The A allele has been associated in vitro with an increased expression of the AGT gene and with higher AGT synthesis. A previous report showed that the G-6A polymorphism of the AGT gene is associated with increased angiotensin production and idiopathic pulmonary fibrosis progression [DOI:10.1183/09031936.00015808]. In the limited number of studies that have been carried out, the AGT M235T variant conferred a risk for developing breast cancer in post-menopausal women. Moreover, ACE plasma levels depend on a 287 bp insertion/deletion (I/D) polymorphism of the ACE gene located on chromosome 17q23. Homozygotes for the D allele have the highest ACE plasma levels, homozygotes for the I allele have the lowest, and ID heterozygotes have intermediate levels. Several studies found an association between ACE I/D polymorphism and lung cancer, albeit some conflicting results [doi:10.1177/1470320314552310]”.

The authors improved the figure 1 and demonstrated the essential production of angiotensin substrates namely the production of renin released from the juxtaglomerular apparatus of the kidney and angiotensinogen from liver cells.

Point 4: The authors further discuss individual molecules of the RAS in the context of cancer hallmarks

  • Proliferation, Invasion, Migration
  • Hypoxia and Angiogenesis
  • Inflammation
  • Tumor Immunological Response

While comprehensively reviewing published molecular finding in these fields, the authors failed to provide an understandable overall picture of these processes, that might help the reader to gain insight. These hallmarks and also the the role of the RAS within these, are critically dependent on the state of disease.

Answer: The authors tried to provide a comprehensive view of the role of RAS in cancer hallmarks adding this information in the Conclusion section.

Conclusion: “Malignant transformation and lung tumour progression is associated with at least six acquired, functional capabilities: sustained angiogenesis, evasion of apoptosis, self-sufficiency in growth signals, insensitivity to anti-growth signals, tissue invasion and metastases, and limitless replicative potential [DOI: 10.1038/nrc2945 ]. The RAS regulates all of these capabilities, although most prominent are its effects on angiogenesis, invasion, pro-survival signalling and proliferation. Inevitably, many of these processes are interdependent and cooperative [DOI: 10.1038/nrc2945 ]. Preclinical studies have provided compelling evidence that the AngII/AT1R axis regulates almost all hallmarks of cancer. It is also clear that AngII/AT1R signaling contributes to the immuno-suppressive tumour microenvironment in multiple ways. The immuno-suppressive milieu is a major barrier for immunotherapy and may explain why immune checkpoint inhibitors failed in NSCLC patients [10.1126/scitranslmed.aan5616]. The high level of expression of the gene encoding the AT1 receptor in the lung tumour suggests that Ang II acting upon the AT1 receptor plays an important role in cell growth, invasion, fibrosis and angiogenesis. The low level and apparent lack of differential expression of the genes encoding components of the ACE2/Ang 1–7/MAS axis in lung tumour tissue promotes the tumorigenic effects of Ang II, while simultaneously reducing the stimulation of the MAS receptor by Ang 1–7, that have contradictory AngII/ATR1 mediated effects [10.1155/2017/6914976]. Multiple clinical studies have also revealed that RAS inhibitors may have beneficial effects in a broad range of malignancies. The gain in survival is tumour type–and stage-dependent and ranged from 3 months (advanced NSCLC) to more than 25 months (metastatic colorectal cancer) in retrospective studies [10.1126/scitranslmed.aan5616]. Finally, the large amount of preclinical and clinical data suggesting a beneficial effect of RAS inhibitors in different cancer types, including lung cancer, we propose that RAS inhibitors have a great potential to become an adjunct within the oncological armamentarium.”

Point 5: Finally, the figures presented by the authors are much too crowded. Trying to combine all aspects discussed previously, Figure S2 shows a very complex network of individual molecular players without transporting a clear message. 

It appears reasonable, considering the complexity of each of the hallmarks discussed, to provide separate Figures for these, which would definitely help not to loose the reader.

Answer: Considering the reviewer suggestion, which we sincerely appreciated, we have simplified manuscript Figure S2, its legend, and the associated information inserted along the text. We hope that now you can find it more comprehensible. Furthermore, we also transformed Figure S2 into Figure 2.1 (Representation of potential angiotensin-associated pathways associated with cell proliferation, invasion, and migration in lung tumour microenvironment); Figure 2.2 (Potential angiotensin-associated pathways involved in hypoxia and angiogenesis in lung tumour microenvironment) and Figure 2.3 (Mechanistic representation of potential angiotensin-associated immunoinflammatory pathways in lung tumour microenvironment).

Once again, we thank you for the time you put in reviewing our paper and look forward to meeting your expectations. Since your inputs have been precious, in the eventuality of a publication, we would like to acknowledge your contribution explicitly.

The authors’

Reviewer 2 Report

Critique

Overall this is a thorough and comprehensive review of the effect of the renin-angiotensin axis in lung pathophysiology. Below are some suggestions to make the manuscript more clear, particularly for individuals not in the RAS or lung cancer fields.

  1. Figure S1 should be in the major text as Fig. 1, not as a supplement only.
  2. Figure S2: see comment #8 below. This figure is complex and overwhelming with utterly too many abbreviations to keep track. Breaking up the pathways into separate figures would be much more instructive and readable.
  3. Line 18 and line 39: Consistency in referring to the axis should be maintained in the manuscript. Either refer to the system as RAS or axis (RAA) but not either/both.
  4. Line 64-73: The ability of chymase to act on angiotensin precursors as well as TGFβ is interesting but the connection and relevance between the two is not evident as written.  Specifically, the text reports two functions of chymase but an argument for the interaction is not made. Rather it appears as two independent functions of chymase. For example, is chymase increased in lung tissue under some circumstances?
  5. Line 82: “interference with cytokine production” is vague. Is “interference” an impairment? If so, is this a downregulation/upregulation of inflammatory cytokines or of anti-inflammatory cytokines?
  6. Line 88-90: this would be better as a statement up above prior to lines 64-73 which would set the stage as to why chymase activity is important and then go on to explain further. As it is, this statement comes late and the relevance of the previous paragraph is not as profound.
  7. Line 100: Would rephrase to emphasize that ACE2 is the receptor for SARS-CoV. “…as a receptor for SARS-CoV and SARS-CoV2.” ACE2 is the receptor; it does not facilitate the receptor.
  8. Line 118-120: Fig. S2 is comprehensive but utterly daunting and complex. For a review (often read by individuals not in the field), separate figures for (a) angiogenesis, (b) metabolism, and (c) migration and invasion functions would be more instructive and communicate the pathways less dauntingly. In fact, it would be better to have the three items a-c above parallel the subheadings in Section 3: (a) cell proliferation (b) hypoxia and angiotenesis, and (c) inflammation. This could then make the figures in the supplement also parallel the text for readers to follow. It would also separate the numerous abbreviations needed for the figures so that readers could follow the pathways more easily, add clarity, and improve the flow and outline of the review.
  9. Line 192ff: Regarding NFAT, the studies in cardiomyocytes have not been found in lung or tumor. If included, there should be explicit statement that the relationship to tumors is speculative based on studies in other tissues.
  10. Lines 222-226: The two statements are contradictory. In fact, injection of tumor into AT2 KO mice showed slower growth in Ref #88.
  11. Iine 250: “axis” should be deleted.
  12. Line 251 and 379: see comment #2 above.
  13. Line 264and line 330: change “blockage” to “blockade”; I may have missed others which should be changed also.
  14. Line 266: if an abbreviation is used once or twice only, eliminate the abbreviation and use only the written full name. Here this revers to MDSC, but should follow this guideline for all abbreviations
  15. Line 278-279: unclear what “it” refers to: Ang II, Ang 1-7, Mas receptor. Define LOX-1.
  16. Section 4.1: It should be noted in the text that none of the trials involved lung cancer. Ref. #125 does not specify how many were lung cancer patients (of the 18 enrolled). Also, they do mention that one lung cancer patient suffered stroke on the Ang (1-7). This should be commented upon.
  17. Line 335: “iACE” should be “ACE inhibitor”; again please avoid new, non-standard, and inconsistent abbreviations.
  18. 4.2: It would be helpful to provide the value of actual risk reduction in the studies mentioned (Refs. 127 - 130) to provide perspective.

Minor grammatical errors should be corrected.

Author Response

Title: Detailed Response to the Editor and to Reviewer 2 for the Manuscript ID: cancers-791140

Firstly, we would like to thank you for your constructive comments in this round of review. Your comments provided valuable insights to refine its contents and analysis. In this document, we try to address the issues raised as best as possible.

  • Figure S1 should be in the major text as Fig. 1, not as a supplement only.

Answer: Figure S1 was changed as Fig 1, being part of the major text.

  • Figure S2: see comment #8 below. This figure is complex and overwhelming with utterly too many abbreviations to keep track. Breaking up the pathways into separate figures would be much more instructive and readable.

Answer: Considering the reviewer suggestion, which we sincerely appreciated, we have simplified manuscript Figure 2, its legend, and the associated information inserted along the text. We hope that now you can find it more comprehensible. Furthermore, we also transformed Figure S2 into Figure 2.1 (Representation of potential angiotensin-associated pathways associated with cell proliferation, invasion, and migration in lung tumour microenvironment); Figure 2.2 (Potential angiotensin-associated pathways involved in hypoxia and angiogenesis in lung tumour microenvironment) and Figure 2.3 (Mechanistic representation of potential angiotensin-associated immunoinflammatory pathways in lung tumour microenvironment).

  • Line 18 and line 39: Consistency in referring to the axis should be maintained in the manuscript. Either refer to the system as RAS or axis (RAA) but not either/both.

Answer: The authors followed the instructions of the reviewer and changed along the manuscript renin-angiotensin axis to renin-angiotensin system (RAS).

  • Line 64-73: The ability of chymase to act on angiotensin precursors as well as TGFβ is interesting but the connection and relevance between the two is not evident as written.  Specifically, the text reports two functions of chymase but an argument for the interaction is not made. Rather it appears as two independent functions of chymase. For example, is chymase increased in lung tissue under some circumstances?

Answer: The authors reformulated the paragraph and tried to explore the chymase's activities for better clarification. In fact, chymase has different activities, that is, on the one hand it is the most efficient angiotensin II-forming enzyme in the human body [doi: 10.1161/CIRCRESAHA.117.310978], on the other hand it performs other functions such as a number of roles in tissue remodeling that are mediated by its direct protease actions: (1) cleavage of matricellular proteins and peptides, including laminin and fibronectin important in cell survival; gap junction proteins essential to regulation of intestinal permeability; and IGF-1 (insulin-derived growth factor 1), which negates the beneficial effects of IGF-1 in ischemia / reperfusion injury; and (2) activation of peptide and enzyme precursors, including MMPs (matrix metalloproteinases) such as MMP-9; TGF-β; stem cell factor; kallikrein, which produces bradykinin and causes further mast cell degranulation and chemotaxis; IL -6 and IL-1β; preproendothelin I; and IL-18, which is involved the pathophysiology of atopic dermatitis [https://doi.org/10.1161/CIRCRESAHA.117.310978]. Chymase has been associated with TGF-β activation and Endothelin-1 formation from preproendothelin-1 in pulmonary fibrosis and chronic obstructive pulmonary disease. [https://doi.org/10.1161/CIRCRESAHA.117.310978]. Chymase activity was noticeably increased in the lungs of animals following bleomycin instillation. A recent report showed prominent accumulation of chymase-expressing mast cells in idiopathic pulmonary arterial hypertension and idiopathic pulmonary fibrosis lungs, and their presence was in close proximity to the regions with marked TGF-β1 expression [10.1183/09031936.00018215]. The role of TGF- β in cancer is further explored in the manuscript.

The proposed paragraph is the following: “Angiotensin I can also be converted to Angiotensin II through the action of chymase, a chymotrypsin-like enzyme that is expressed in the secretory granules of mast cells [16]. In fact, chymase is the most efficient angiotensin II-forming enzyme in the human body and has been implicated in a wide variety of human diseases that also implicate its many other protease actions: (1) cleavage of matricellular proteins and peptides, including laminin and fibronectin important in cell survival; (2) activation of peptide and enzyme precursors, including matrix metalloproteinases (MMPs) such as MMP-9, TGF-β, stem cell factor, kallikrein, IL -6 and IL-1β, preproendothelin I and IL-18 [doi: 10.1161/CIRCRESAHA.117.310978]. Chymase activity was noticeably increased in the lungs of animals following bleomycin instillation. A recent report showed prominent accumulation of chymase-expressing mast cells in idiopathic pulmonary arterial hypertension and idiopathic pulmonary fibrosis lungs, and their presence was in close proximity to the regions with marked TGF-β1 expression [10.1183/09031936.00018215]. Furthermore, tumour sections from 32 cases of adenocarcinoma and 13 cases of squamous cell carcinoma revealed significantly higher counts of chymase-positive mast cells in the central region than those in the normal regions [DOI:10.1016/j.ejcts.2005.06.020]. Chymase has been associated with TGF-β activation and Endothelin-1 formation from preproendothelin-1 in pulmonary fibrosis and chronic obstructive pulmonary disease [https://doi.org/10.1161/CIRCRESAHA.117.310978].

  • Line 82: “interference with cytokine production” is vague. Is “interference” an impairment? If so, is this a downregulation/upregulation of inflammatory cytokines or of anti-inflammatory cytokines?

Substantial evidence indicates that Ang II triggers inflammatory process in multiple tissues and cells mainly through the activation of AT1R. Plasma and lung Ang II levels were dramatically increased in a time-dependent manner in models of rat acute lung injury. These results are consistent with studies in which the AT1R antagonist, losartan, prevented acute lung injury-induced interstitial edema and inflammatory cell infiltration [doi: 10.1097/SHK.0000000000000633]. Moreover, Ang II induces differentiation of immune cells and augmentation of pro-inflammatory cytokine production, both contributing to elevated blood pressure and sustained hypertensive conditions [doi: 10.1007/s11906-018-0900-0]. 

Thus, the authors reformulated the statement: “Upon activation of this receptor, a pleiotropic regulatory role is induced at the lung, resulting in vasoconstriction, cell proliferation, angiogenesis, and augmentation of pro-inflammatory cytokine production, inflammatory cell chemotaxis, epithelial cell apoptosis, increased oxidative stress and fibrosis”.

  • Line 88-90: this would be better as a statement up above prior to lines 64-73 which would set the stage as to why chymase activity is important and then go on to explain further. As it is, this statement comes late and the relevance of the previous paragraph is not as profound.

Answer: The authors followed the instructions of the reviewer and changed the statement “Therefore, local upregulation of the expression of either angiotensin II or TGF-β1, which might influence each other’s activity or synergy, should be blocked in order to delay the progression of lung fibrotic disease” prior to the description of chymase activity.

  • Line 100: Would rephrase to emphasize that ACE2 is the receptor for SARS-CoV. “…as a receptor for SARS-CoV and SARS-CoV2.” ACE2 is the receptor; it does not facilitate the receptor.

Answer: The authors reformulated the statement: “ACE2 has a multiplicity of physiological roles that revolve around its trivalent function: a negative regulator of the RAS, facilitator of amino acid transports, and is the entry receptor for SARS-CoV and SARS-CoV-2”.

  • Line 118-120: Fig. S2 is comprehensive but utterly daunting and complex. For a review (often read by individuals not in the field), separate figures for (a) angiogenesis, (b) metabolism, and (c) migration and invasion functions would be more instructive and communicate the pathways less dauntingly. In fact, it would be better to have the three items a-c above parallel the subheadings in Section 3: (a) cell proliferation (b) hypoxia and angiotenesis, and (c) inflammation. This could then make the figures in the supplement also parallel the text for readers to follow. It would also separate the numerous abbreviations needed for the figures so that readers could follow the pathways more easily, add clarity, and improve the flow and outline of the review.

Answer: Considering the reviewer suggestion, which we sincerely appreciated, we have simplified manuscript Figure 2, its legend, and the associated information inserted along the text. We hope that now you can find it more comprehensible. Furthermore, we also transformed Figure S2 into Figure 2.1 (Representation of potential angiotensin-associated pathways associated with cell proliferation, invasion, and migration in lung tumour microenvironment); Figure 2.2 (Potential angiotensin-associated pathways involved in hypoxia and angiogenesis in lung tumour microenvironment) and Figure 2.3 (Mechanistic representation of potential angiotensin-associated immunoinflammatory pathways in lung tumour microenvironment).

  • Line 192ff: Regarding NFAT, the studies in cardiomyocytes have not been found in lung or tumor. If included, there should be explicit statement that the relationship to tumors is speculative based on studies in other tissues.

Accumulating studies have suggested that NFATs are involved in many aspects of cancer, including carcinogenesis, cancer cell proliferation, metastasis, drug resistance and tumour microenvironment [https://doi.org/10.1016/j.canlet.2015.03.005].

However, the crosstalk between Ang-(1-7) and NFAT in lung cancer has never been explored. As such, the authors reformulated the statement: “Further data from cardiomyocytes supports a nitric oxide/guanosine 3’,5’-cyclic monophosphate-dependent pathway, which modulated the activity of the transcription factor NFAT (nuclear factor of activated T-cells), preventing its translocation to the nucleus [73], where is known to transcriptional regulate genes involved in malignant cell proliferation, migration and metastasis [74]. However, the crosstalk between ACE2/Ang-(1-7)/Mas axis and NFAT upregulation in lung cancer has never been explored and the relationship to tumours is speculative based on studies in other tissues.”

  • Lines 222-226: The two statements are contradictory. In fact, injection of tumor into AT2 KO mice showed slower growth in Ref #88.

The authors corrected the statement and reformulated: “Ex vivo experiments in Lewis lung tumour xenografts revealed a significant decrease in angiogenesis after inhibition of AT2 receptor or in AT2 receptor-knockout mice with impaired production of proangiogenic factors included VEGF (88).”

  • Iine 250: “axis” should be deleted.

Answer: The authors deleted the term “axis”.

  • Line 251 and 379: see comment #2 above.
  • Line 264and line 330: change “blockage” to “blockade”; I may have missed others which should be changed also.

Answer: The authors performed the correction “blockage” to “blockade” along the manuscript.

  • Line 266: if an abbreviation is used once or twice only, eliminate the abbreviation and use only the written full name. Here this revers to MDSC, but should follow this guideline for all abbreviations

Answer: The authors followed the instructions of the reviewer and eliminated the unnecessary abbreviations, including MDSC.

  • Line 278-279: unclear what “it” refers to: Ang II, Ang 1-7, Mas receptor. Define LOX-1.

Answer: The authors reformulated and replaced “it” to Ang-(1-7) as well as defined LOX-1: “Conversely from AT1 receptor activation by Ang II, the Ang-(1-7) binding to Mas receptor triggers a down regulation of pro-oxidant pathways, preventing or attenuating the oxidative stress-induced cellular damage (98), although Ang-(1-7) can also inhibit Ang II-activated inflammation through a deregulatory effect in lectin-like oxidized low-density lipoprotein receptor-1.”

  • Section 4.1: It should be noted in the text that none of the trials involved lung cancer. Ref. #125 does not specify how many were lung cancer patients (of the 18 enrolled). Also, they do mention that one lung cancer patient suffered stroke on the Ang (1-7). This should be commented upon.

Answer: The authors did not refer to table S1 where the main clinical trials on RAS and cancer are depicted. Two trials were performed in lung cancer patients and consisted on the study of the effect of ACE inhibitors in NSCLC patients previously treated with radiation therapy. This is depicted in table S1. The authors improved this section and describe the relevant clinical trials and discuss their results. Concerning the phase II clinical trial (ref.#125), it was only included 1 patient with lung cancer. Furthermore, one patient developed toxicity grade 4 (stroke) and reversible cranial neuropathy (grade 3). The authors consider this information relevant and included in this section. Therefore, the authors reformulated section 4.1: “Despite basic and pre-clinical evidence concerning the impact of RAS in cancer hallmarks, the information available regarding clinical trials remains scarce, particularly in lung cancer. The primary objective of a Phase II Randomized Trial (NCT00077064, table S1) was to test the ability of captopril to alter the incidence of pulmonary damage at 12 months after completion of radiation in patients with non-small cell lung cancer or limited-stage small cell lung cancer. Due to the unmet accrual/randomization goals, it was not possible to adequately test the hypothesis that captopril ameliorates radiotherapy-induced pulmonary toxicity. Another double-blind placebo-controlled randomized trial (NCT01880528, table S1) enrolled 23 patients with lung cancer to study the putative protective effect of lisinopril in pneumonitis induced by radiotherapy. The results of this clinical trial suggest that there was a clinical signal for safety and possibly beneficial in concurrently administering lisinopril during thoracic radiotherapy to mitigate or prevent radiation-induced pulmonary distress [doi: 10.1016/j.ijrobp.2018.10.035]. However, larger-scale randomized phase 3 trials are needed in the future to confirm these results. Notably, a multicenter clinical trial showed that losartan stabilized lung function in patients with idiopathic pulmonary fibrosis over 12 months (NCT00879879, table S1), reinforcing the importance of AT1 receptor blockers in the impairment of fibrosis progression. Moreover, the treatment with AT1 receptor blockers might potentiate the response to immunotherapy in lung cancer patients due to a hypothetic impairment of a fibrotic immunosuppressive microenvironment. Nevertheless, there is no study to prove this hypothesis. A small Phase I study (NCT00471562, table S1) of Ang-(1–7) as a first-in-class anti-angiogenic drug targeting Mas receptor, was undertaken in 18 patients with advanced solid tumours refractory to standard therapy with only one patient with lung cancer included [125]. Dose-limiting toxicities encountered at the 700 μg/kg dose included stroke (grade 4) and reversible cranial neuropathy (grade 3). Other toxicities were generally mild. Clinical benefit, defined as disease stabilization for more than three months, was observed in two of the three patients with metastatic sarcoma and it was associated with reduction of PlGF plasma levels [125]. A Phase II study failed to confirm PlGF as biomarker of response to Ang-(1-7) administration [126]. However, two patients with vascular sarcomas demonstrated prolonged disease stabilization of 10 months (hemangiopericytoma) and 19 months (epithelioid hemangioendothelioma) under Ang-(1-7) treatment, thereby requiring further investigation [126].”

  • Line 335: “iACE” should be “ACE inhibitor”; again please avoid new, non-standard, and inconsistent abbreviations.

Answer: The authors changed “iACE” to “ACE inhibitor” along the manuscript and deleted all new or inconsistent abbreviations.

  • 4.2: It would be helpful to provide the value of actual risk reduction in the studies mentioned (Refs. 127 - 130) to provide perspective.

Most of the retrospective studies and meta-analysis describe a protective effect of RAS blockers (ACE inhibitors and AT1 receptor antagonists) in NSCLC patients, regardless the stage. Nevertheless, a recent large study population-based cohort that used ACE inhibitors suggested an increased risk for developing lung cancer, particularly among subjects with more than five years under treatment. Nevertheless, the risk for lung cancer was not confirmed in patients under AT1 receptor antagonists [129]. These findings might rely on ACE inhibition-associated accumulation of bradykinin in the lung, rather than a decrease of angiotensin II signalling, especially when higher dosages of ACE inhibitors are taken into consideration. As such, the authors concluded with the following statement: “When analysed together, these results from clinical and pre-clinical studies suggest that AT1 receptor blockers, in contrast to ACE inhibitors, might be putative therapeutic tools to impair lung cancer progression. Also, compelling evidence supported that AT1 receptor blockers are significantly associated with reduced lung cancer risk.”

Once again, we thank you for the time you put in reviewing our paper and look forward to meeting your expectations. Since your inputs have been precious, in the eventuality of a publication, we would like to acknowledge your contribution explicitly.

The authors’

Reviewer 3 Report

Brief summary 

In this review, the authors provide an overview of the role of the renin-angiotensin-aldosterone system (RAS) in tumor cell malignancy and tumor microenvironment modulation during cancer progression, with particular focus on lung cancer. By first explaining the enzymatic activity and distribution of different members of the RAS in the lung tissue, they then focus on the involvement of RAS in different sequential aspects of cancer development (hallmarks of cancer). The presented literature points to different member of the RAS having either pro- or anti-tumorigenic/metastatic properties, providing several possible targets for cancer therapy. Finally, the authors provide an overview of the current clinical evidence of the effect of RAS modulators in cancer patients, suggesting that clinical trials are needed to conclusively assess the effect of RAS therapeutics on lung cancer incidence and progression.

Broad comments 

Areas of strength:

  • This novelty of this review is to specifically review the existing literature on the role of the RAS in lung tissue and lung cancer, independent of the canonical systemic RAS, highlighting the potential benefit of RAS therapy specifically in this disease. Previous reviews have addressed the role of RAS in other malignancies but did not focus specifically on lung cancer.
  • The role of the RAS in different aspects of cancer progression and tumor-microenvironment crosstalk have been extensively reviewed by the authors. In particular, the opposite roles of the ACE/AngII/AT1 receptor and ACE2/Ang(1-7) Mas receptor axis in different aspects of cancer development are nicely detailed.
  • The supplementary figures accompanying the main text are very comprehensive and well depicted.

Areas of weakness:

  • In this review the authors have omitted to explain the basics of the signaling and physiological significance of the RAS. For example, the general role of angiotensin peptides, which are the main focus of the review, is not introduced. In section 2 references to other reviews have been made, however a general overview of the system is needed to make the rest of the review more easily understandable by non-expert readers.
  • Despite providing a complete description of the synthesis and signaling of angiotensin peptides in the lung tissue under physiological conditions (section 2), the authors omit to explain the distribution of those enzymes and receptors in the cancer setting. It would be interesting to know the expression of RAS components during lung cancer progression, for example from biopsies, and the known mutations that affect these factors in (lung) cancer patients.
  • Some of the literature described in this review does not involve lung cancer. In particular, the role of RAS in cell proliferation and invasion (section 3.1) and anti-tumor immunity (section 3.4) has been largely studied in other types of cancer/cells, while evidence in lung cancer models seems to be tenuous. Although pre-clinical studies suggest that ACE/AngII/AT1 blockade and ACE2/Ang(1-7)/Mas axis stimulation can reprogram lung cancer microenvironment and limit tumor growth, the rationale for RAS modulation specifically in lung cancer is not clearly stated.
  • Similarly, as stated by the authors, clinical trials addressing the role of RAS therapeutics in lung cancer incidence and progression have been lacking (section 4). The authors briefly mention (although without much detail) that some potential difficulties might complicate the introduction of RAS modulators in clinical settings (section 5). These observations are very important and might highlight potential pitfalls in the introduction of RAS therapeutics in the treatment of lung cancer. Such aspects, however, are not sufficiently addressed by the authors.
  • Some sections are excessively information-dense with little background on the signaling pathways described, for example sections 3.1. Readers might not be familiar with all of the signaling pathways reported. Authors might want to consider simplifying some of these sentences to make the main concepts more easily understandable. Instead of a list of previously published evidence, the authors should convey more general concepts and then delve into some details.
  • Some sections are overly simplified. For example, in section 3.3 the authors describe the role of RAS components in inflammation and focus exclusively on macrophages. However, the effect of RAS on the immune systems (in particular myeloid cells) goes far beyond the modulation of the inflammatory response. For instance, RAS components are also expressed in other immune cells such as neutrophils and dendritic cells. A broader view of the role of RAS in the innate immune system would be desirable.
  • Despite being beautifully comprehensive and designed, Figure S2 is far more detailed than the text referring to it. A simpler figure summarizing the main concepts described in the review might be more powerful at conveying the intended message.

Specific comments 

Scientific comments (Major comments are specified):

MAJOR - Line 2 (Title) – The authors might consider revising the current title. “lung tumor and microenvironment interaction” is not clear. Also, throughout the review the authors refer to the “renin-angiotensin system” instead of “renin-angiotensin axis” as in the title and abstract. Please consolidate for consistency.

Line 18 and 34 – Clarify the difference between RAA (in the abstract) and RAS (throughout the text) and, if synonym, use one or the other definition throughout the text for consistency.

Line 21 – Please change/remove “In light of compelling evidence”.

Line 24-25 – Consider rephrasing “clinical intervention of this pathway”  to “blockade of this pathway in clinical settings”.

MAJOR - Line 43-44 – Here the authors should explain the physiological RAS hormone system/signaling much more into details, instead of referencing to previous reviews on the topic, in order to make the rest of the review more easily understandable to non-expert readers. In particular, the physiological role of angiotensin peptides should be explained here.

MAJOR - Lines 53-55 – Here, the authors make the point that intra-pulmonary RAS is involved in the etiology of lung diseases. I feel that this point should be made clearer/stronger, followed by the list of experimental evidence that the authors nicely summarize in lines 57-116.

Line 77-78 – AGTR1 and AGTR2 should be defined here. What proteins do they encode?

Line 85 – Transition of what cells?

Line 88 – What does local refer to? I guess lungs, but please explain.

MAJOR - Lines 43-116 (section 2) – In section 3, the authors describe that a variety of pro- or anti-tumor effects of angiotensin peptides rely on their signaling through AT1 and AT2 on tumor cells. In section 2 or a separate section the authors should focus on the expression pattern of AT1 and AT2 receptor in cancer cell lines, patient biopsies and tumor microenvironment of the lungs. Some of these pieces of information are scattered throughout the text and should be combined in the same section. A paragraph on the known mutations affecting these factors in cancer would also be very informative.

Line 118 – RAS is not a mechanism but a signaling system. Please rephrase.

Line 120 – “Metabolism” is not a tumor-associated phenotype, please rephrase.

Line 121 and section 3.1. - Tumor cell proliferation and metastasis/invasion/migration are two distinct processes in cancer progression. If possible, the authors should attempt to address these two aspects in separate subsections of chapter 3.

Line 144-146.- This sentence seems to be unrelated to the previous one. Consider moving this somewhere else.

Line 149 – “Neo-vessel” is not a conventional scientific terminology. Please change to “new vessels” or “neovascularization”.

Line 150 – The role of CSC in cancer metastasis is still a matter of debate and the mechanisms are not yet completely uncovered. The authors should rephrase the expression”compelling evidence demonstrating...” to make it less decisive.

Line 151-155 – Although ang II/ATR have a role in EMT, EMT was not listed among mechanisms of cancer metastasis in the previous sentence. Please expand.

Line 157 – Does AGTR1 refer to the gene (AGTR1, in italic) or the protein (ATR1)?

Line 158 – A lot of the abbreviations in this sentence (CXCR4/SDF-1α signaling through  FAK/RhoA/ROCK1/2/MLC) should be expanded and/or omitted for clarity.

MAJOR - Line 123-143 – Despite the sentence on lines 159-161, the pro-proliferative and pro-metastatic roles of Angiotensin II/ATR1 are general and, in most cases, do not appear to have been demonstrated in lung cancer models, either in vitro or in vivo. As highlighted in figure 2, these pathways might “potentially” apply to lung cancer. The authors should highlight the relevance of these mechanisms to lung cancer, if any. Similar considerations need to be made for section 3.1.2 and 3.1.3.

Line 196-197 – Evidence presented by the authors supports the idea that increased ACE2 activity and Ang(1-7) levels have anti-tumor and anti-metastasis effects. In this sense “targeting” should be rephrased to emphasize that the overexpression, and not the blockade, of these proteins might be beneficial in the context of lung cancer. Comments should also be made in regards to what strategy might be more effective to induce the activation of the ACE2/Ang(1-7)/Mas axis in patients, if any.

MAJOR - Line 198 -  A notable omission of the authors is the paper: Attoub S, Gaben AM, Al-Salam S, et al. Captopril as a potential inhibitor of lung tumor growth and metastasis. Ann N Y Acad Sci. 2008;1138:65‐72. doi:10.1196/annals.1414.011, which provides pre-clinical data of the effect of captopril on lung tumor growth and metastasis.

Lines 200-207 – This paragraph should introduce the subsections 3.2.1-2. Experimental data regarding the role of AngII, ACE and AT1 in hypoxia and angiogenesis should be moved to the following subparagraphs.

Line 206 – RAS is a system and it is not “expressed”. Please rephrase.

Line 214 – Expand on candesartan pharmacological action (e.g. target?), for clarity. Non-experts are not familiar with this drug.

Line 215 and 218-219 – “...mRNA and protein...” and “...at both mRNA and protein level” are details not needed here. “Expression” should suffice.

MAJOR - Line 236 – It is clear from other reviews that the effect of RAS components on the immune system goes beyond modulation of the inflammatory response. RAS components are also expressed in neutrophils and dendritic cells. What is the role of these myeloid cells in lung cancer etiology and how does the RAS affect it? Authors should expand on the role of RAS in the innate immune system of lung cancer beyond inflammation.

Line 290 – Inflammation is a process and cannot be upregulated. Please rephrase.

MAJOR - Line 306-326 – Similarly to line 123-143, the studies reported here were done in cancer models different to lung cancers. Authors should address the relevance of this pathways to lung cancer, if any. Also, for consistency with the other subsection of section 3, also this paragraph should be formatted as “ACE/Ang II/AT1 receptor axis and AT2 receptor” and “ACE2/Ang(1-7)/Mas receptor axis”, if possible.

MAJOR - Line 333 and section 4 – A major omission of the authors is the paper “Aydiner, Adnan MD; Ciftci, Rumeysa MD; Sen, Fatma MD Renin-Angiotensin System Blockers May Prolong Survival of Metastatic Non-Small Cell Lung Cancer Patients Receiving Erlotinib, Medicine: June 2015 - Volume 94 - Issue 22 - p e887 doi:10.1097/MD.0000000000000887”, where RAS blockers in combination with erlotinib improve overall survival of NSCLC patients.

MAJOR - Lines 334-343 – Considering the lack of clinical trials on lung cancer, which is the main focus of this review, I would consider omitting the subsection 4.1, unify subsection 4.2 with 4 and add the sentences of lines 334-337 to the newly unifies section.

Line 352 – Please rephrase “adjuvant specific biological mechanism of action” for clarity.

Line 356-357 – Is it an increased or decreased risk of lung cancer that was not confirmed in patients under ARBs? Please correct sentence.

MAJOR - Line 371-374 – The sentence “The major challenge in clinical context is the complex nature of RAS signalling, which seems to be variable in response to several factors, making the response to RAS therapeutics, either individually or in combination with other drugs, difficult to predict” is particular important for the purpose of this review. I suggest that the authors overview these aspects into details under the section “Clinical trials”.

Figure S2 – This figure has far more molecular details that the text referring to it (sections 3). Figure should be simplified to reflect the information that is written in the main body of the review.

Figures S1 and S2 – Both figures should be moved to the main text, upon simplification (Figure 2) and shortening of the figure legends.

Language comments:

Lines 18-25 (Abstract) and all text - Consider rephrasing the expressions “supporting/recent/compelling evidence” or “previous studies” with other forms (e.g. first author of cited paper) to avoid repetition or removing them to make reading more fluid. In particular, the abstract should be reviewed to avoid repetition of the word “evidence”.

Lines 29-31 – Review sentence to avoid repetition of “worldwide”.

Line 34 - The abbreviation RAS for “renin-angiotensin-aldosterone system” should be defined here, where it appears for the first time.

Line 35, 222 and 370 – “Notwithstanding” is not commonly used in English, consider changing to “Nevertheless”.

Line 37 – “,thereby critically” is incorrect here, please rephrase.

Line 38 – Please change “updated information”.

Line 44-45 – The definition of RAS should be introduced before and only the abbreviation used here.

Line 46-49 – consider rephrasing to avoid repetition of “local”.

Line 51 – “On its turn” is not a correct English expression and should be changed to “In turn” or “subsequently”, although the exact connector meant by the authors is not clear here.

Line 53 – “In fact” is not used correctly here, as this sentence does not explain the previous one. Please rephrase or remove.

Line 55-56 – Rephrase “Add to knowledge supporting...”.

Line 74, 94, 122, 125, 130, 141, 151 and throughout the text– “Ang” should be expanded to angiotensin for consistency or, alternatively, angiotensin should be abbreviated to “ang” throughout the text.

Line 89-90 – Please rephrase “which might influence each other’s activity or synergy” for clarity.

Line 114 – “having as surrogate the hallmarks of cancer” is incorrect, please revise.

Line 126-130 – Consider divinding this sentence in 2 shorter ones.

Line 136 – Change “is incompletely understood” to “is not completely understood”. “Support a role ... as synergistically efficacious” is not correct, please rephrase to “show an effect” or similar.

Line 142  - Rephrase “all intracellular signaling pathways”.

Line 147 – Change “related” with “associated” or similar.

Line 171 – “...among..” is used incorrectly here. Please correct. “Signaling” is US English. Considering that the rest of the review has been formatted in UK English, please change to “signalling”.

Line 178-179 – “Some studies demonstrated that...” and “...have been associated with” are incompatible in the same sentence. Please rephrase.

Line 183 – The abbreviation “COX” needs to be expanded to “cyclooxygenase (COX)...”.

Line 184-185 – “By an effect to under-express...” is incorrect. Please rephrase. “Under-express” and “under-activate” are not conventional scientific language, please revise.

Line 189 – “in A breast cancer model”.

Line 191 – Correct “e-cadherin” to “E-cadherin”

Line 194 – “where is known to transcriptional regulate” is incorrect and needs rephrasing.

Line 205 – “...previous studies demonstrated that...” is not needed here.

Line 220 – Reference  [84] should be moved to the end of the sentence.

Line 222 – “Notwithstanding” is not correct here, as there is no connection with the previous sentence. Consider changing to “In contrast to AT1...” or similar.

Line 238 – Consider changing “major precursors”.

Line 239 – “...microenvironment characteristics” cannot be rich in something, but tumor microenvironment can. Please revise.

Line 240 – “cell-proliferation” is pleonastic, please remove.

Line 241 – Rephrase to “Inflammation has been postulated to play a key role in lung carcinogenesis” or similar.

Line 243 – Use either “et al.” or “and colleagues” throughout the text, for consistency.

Line 245-247 – The sentence “exposure of agents...inflammation” is incorrect. Please revise.

Line 251 – Change “RA” to “RAS”.

Line 253 – “Intensely” is not correctly used here, please change.

Line 264 – “blockage” should be substituted with “blockade”.

Line 270 – Please rephrase “Despite conflicting data, Ang II/AT1 receptor signalling stimulates...”.

Line 273 –  “...albeit, taken together” is incorrect. Consider starting a new sentence and rephrase.

Line 276 – “Conversely from” is incorrect, please change.

Line 279 – “cumulatively” is not used correctly here. Please remove.

Line 283 – Change “in macrophage function” to “on macrophage function”?

Line 319 – Please correct “this result led to...”.

Line 325 – The phrase “get into further detailed knowledge regarding RAS through experimental and clinical research in lung cancer and its influence in immunotherapy response” needs rephrasing.

Line 339-340 – If PlGF is equal to PLGF previously indicated in the text, please format similarly.

Line 353 – “supported” is not correct here, please change to “showed” or similar.

Line 355 – A large cohort cannot “suggest”, please rephrase.

Line 356 – Please correct “subjects with more than five years under treatment”.

Line 367 – Consistency is needed throughout the text for the terms “in vivo” and “in vitro” in terms of italics font. The use of “” here is not needed.

Line 377 – Please correct the typo “oppening”.

Line 379 and 382 – Consider inverting “Finally” and “Moreover”.

Author Response

Title: Detailed Response to the Editor and to Reviewer 3 for the Manuscript ID: cancers-791140

Firstly, we would like to thank you for your constructive comments in this round of review. Your comments provided valuable insights to refine the manuscript contents and analysis. In this document, we try to address the issues raised as best as possible.

Areas of strength:

  • This novelty of this review is to specifically review the existing literature on the role of the RAS in lung tissue and lung cancer, independent of the canonical systemic RAS, highlighting the potential benefit of RAS therapy specifically in this disease. Previous reviews have addressed the role of RAS in other malignancies but did not focus specifically on lung cancer.
  • The role of the RAS in different aspects of cancer progression and tumor-microenvironment crosstalk have been extensively reviewed by the authors. In particular, the opposite roles of the ACE/AngII/AT1 receptor and ACE2/Ang(1-7) Mas receptor axis in different aspects of cancer development are nicely detailed.
  • The supplementary figures accompanying the main text are very comprehensive and well depicted.

Thank you for your comment.

Areas of weakness:

  • In this review the authors have omitted to explain the basics of the signaling and physiological significance of the RAS. For example, the general role of angiotensin peptides, which are the main focus of the review, is not introduced. In section 2 references to other reviews have been made, however a general overview of the system is needed to make the rest of the review more easily understandable by non-expert readers.

The authors would like to thank you for your pertinent comment. We understand that an overview of the normal function of the system, as well as ramifications of its dysfunction is important to describe. However, the authors chose to simplify and explain the RAS through figure 1. Furthermore, it is widely recognized that this classical view of the endocrine RAS pathway represents an incomplete description of the system. Instead of one simple circulating RAS, it is recognized that there are also several tissue (local) renin-angiotensin systems that function independently of each other and of the circulating RAS. In particular, angiotensin II generation at the tissue level by these local systems appears to have physiologic effects that are as important as circulating angiotensin II and, under some circumstances, even more important. Thus, the authors explored the pulmonary RAS, although we reformulated the section 2 in order to provide an overview of RAS.

Line 43-44: “The RAS was first discovered and studied in the physiological regulation of blood pressure, fluid homeostasis and pathogenesis of hypertension (1). Angiotensin II (AngII), the final effector of the system, causes vasoconstriction both directly and indirectly by stimulating Ang II type 1 receptor (AT1 receptor) present on the vasculature and by increasing sympathetic tone and arginine vasopressin release. Chronically, Ang II regulates blood pressure by modulating renal sodium and water reabsorption directly, by stimulating AT1 receptors in the kidney, or indirectly, by stimulating the production and release of aldosterone from the adrenal glands, or stimulating the sensation of thirst in the central nervous system (https://doi.org/10.1210/en.2003-0150). The enzymatic cascade by which Ang II is produced consists of renin, which cleaves angiotensinogen (AGT) to form the decapeptide angiotensin I. Ang I is then further cleaved by angiotensin-converting enzyme (ACE) to produce the octapeptide Ang II, the physiologically active component of the system. Further degradation by aminopeptidase A and N produces angiotensin III (Ang 2–8), and angiotensin IV (Ang 3–8), respectively (Figure 1) (https://doi.org/10.1210/en.2003-0150). The actions of Ang II result from its binding to specific receptors (AT1 and AT2). Ang II, via its AT1 receptor, is also involved in cell proliferation, left ventricular hypertrophy, nephrosclerosis, vascular media hypertrophy, endothelial dysfunction, neointima formation and processes leading to athero-thrombosis (DOI: 10.1080/08037050310001057). Ang II, via its AT2 receptor, is involved in cell differentiation, tissue repair, apoptosis and vasodilation (DOI: 10.1080/08037050310001057). Other receptors have been described in relation to the RAS. For instance, an AT4 receptor binding site has been identified, and in contrast to the other AT receptors, it does not seem to be a G protein-coupled receptor (doi: 10.1016/0167-0115(92)90527-2). This receptor binds Ang IV (3–8) preferentially, has been localized in various mammalian tissues, and suggested to cause vasodilatation (doi: 10.1016/0167-0115(92)90527-2).”

  • Despite providing a complete description of the synthesis and signaling of angiotensin peptides in the lung tissue under physiological conditions (section 2), the authors omit to explain the distribution of those enzymes and receptors in the cancer setting. It would be interesting to know the expression of RAS components during lung cancer progression, for example from biopsies, and the known mutations that affect these factors in (lung) cancer patients.

The authors reformulated section 2 and described the distribution of RAS components in lung adenocarcinoma. Moreover, the authors also explored the role of the circulating RAS deregulation in lung diseases including lung cancer and described the known germline mutations that might affect RAS components.

Line 80: “The components of RAS are frequently differentially expressed in various cancers including brain, lung, pancreatic, breast, prostate, colon, skin and cervical carcinomas in comparison with their corresponding non-malignant tissue (doi:10.1093/carcin/bgn171). A large-scale study of estrogen receptor-positive breast cancer tumours revealed an increase in AGTR1 mRNA expression. Contradictory results were observed in lung adenocarcinoma tissues since AGTR1, AGTR2 as well as ACE mRNA expression were underexpressed. Notably, AGT was overexpressed in lung adenocarcinoma tissue. (PMID 28791183, https://doi.org/10.1371/journal.pone.0226553).”  

Line 114: “The circulating RAS might also be important in providing the substrates for local modulation of angiotensin peptides with influence in lung diseases. Genetic and epidemiological evidence provides support that germline mutations in RAS components contribute to the risk of developing idiopathic pulmonary fibrosis, acute lung injury and certain malignancies. For example, the human gene encoding angiotensinogen (AGT) is located on chromosome 1q42.3 and a large number of single-nucleotide polymorphisms (SNPs) have been described. Among them, the SNPG-6A nucleotide substitution at position 6, upstream from the initial transcription start codon, has been studied particularly. The A allele has been associated in vitro with an increased expression of the AGT gene and with higher AGT synthesis. A previous report showed that the G-6A polymorphism of the AGT gene is associated with increased angiotensin production and idiopathic pulmonary fibrosis progression [DOI:10.1183/09031936.00015808]. In the limited number of studies that have been carried out, the AGT M235T variant conferred a risk for developing breast cancer in post-menopausal women. Moreover, ACE plasma levels depend on a 287 bp insertion/deletion (I/D) polymorphism of the ACE gene located on chromosome 17q23. Homozygotes for the D allele have the highest ACE plasma levels, while homozygotes for the I allele have the lowest, and ID heterozygotes have intermediate levels. Several studies found an association between ACE I/D polymorphism and lung cancer, albeit some conflicting results [doi:10.1177/1470320314552310]”.

  • Some of the literature described in this review does not involve lung cancer. In particular, the role of RAS in cell proliferation and invasion (section 3.1) and anti-tumor immunity (section 3.4) has been largely studied in other types of cancer/cells, while evidence in lung cancer models seems to be tenuous. Although pre-clinical studies suggest that ACE/AngII/AT1 blockade and ACE2/Ang(1-7)/Mas axis stimulation can reprogram lung cancer microenvironment and limit tumor growth, the rationale for RAS modulation specifically in lung cancer is not clearly stated.

There are limited studies examining the Ang II-mediated signaling pathways activated in lung cancer cells concerning cell proliferation and invasion as well as lung cancer models to study the role of RAS in anti-lung tumour immunity. The authors explored the most relevant studies available in the literature about lung cancer. In fact, there are more in vitro and in vivo studies on other cancer models that explore the role of RAS in cell proliferation and immune response, within the tumour microenvironment. Although not described in lung cancer, the authors consider that the inclusion of further studies in other models of cancer allows transmitting, in a more comprehensive way, the role of the different components of RAS in cancer. These signaling pathways found in other cancer models may also be common in lung cancer.

  • Similarly, as stated by the authors, clinical trials addressing the role of RAS therapeutics in lung cancer incidence and progression have been lacking (section 4). The authors briefly mention (although without much detail) that some potential difficulties might complicate the introduction of RAS modulators in clinical settings (section 5). These observations are very important and might highlight potential pitfalls in the introduction of RAS therapeutics in the treatment of lung cancer. Such aspects, however, are not sufficiently addressed by the authors.

The authors improved this section and described the relevant clinical trials and discuss their results. Therefore, the authors reformulated section 4.1: “Despite basic and pre-clinical evidence concerning the impact of RAS in cancer hallmarks, the information available regarding clinical trials remains scarce, particularly in lung cancer. The primary objective of a Phase II Randomized Trial (NCT00077064, table S1) was to test the ability of captopril to alter the incidence of pulmonary damage at 12 months after completion of radiation in patients with non-small cell lung cancer or limited-stage small cell lung cancer. Due to the unmet accrual/randomization goals, it was not possible to adequately test the hypothesis that captopril ameliorates radiotherapy-induced pulmonary toxicity. Another double-blind placebo-controlled randomized trial (NCT01880528, table S1) enrolled 23 patients with lung cancer to study the putative protective effect of lisinopril in pneumonitis induced by radiotherapy. The results of this clinical trial suggest that there was a clinical signal for safety and possibly beneficial in concurrently administering lisinopril during thoracic radiotherapy to mitigate or prevent radiation-induced pulmonary distress [doi: 10.1016/j.ijrobp.2018.10.035]. However, larger-scale randomized phase 3 trials are needed in the future to confirm these results. Notably, a multicenter clinical trial showed that losartan stabilized lung function in patients with idiopathic pulmonary fibrosis over 12 months (NCT00879879, table S1), reinforcing the importance of AT1 receptor blockers in the impairment of fibrosis progression. Moreover, the treatment with AT1 receptor blockers might potentiate the response to immunotherapy in lung cancer patients due to a hypothetic impairment of a fibrotic immunosuppressive microenvironment. Nevertheless, there is no study to prove this hypothesis. A small Phase I study (NCT00471562, table S1) of Ang-(1–7) as a first-in-class anti-angiogenic drug targeting Mas receptor, was undertaken in 18 patients with advanced solid tumours refractory to standard therapy with only one patient with lung cancer included [125]. Dose-limiting toxicities encountered at the 700 μg/kg dose included stroke (grade 4) and reversible cranial neuropathy (grade 3). Other toxicities were generally mild. Clinical benefit, defined as disease stabilization for more than three months, was observed in two of the three patients with metastatic sarcoma and it was associated with reduction of PlGF plasma levels [125]. A Phase II study failed to confirm PlGF as biomarker of response to Ang-(1-7) administration [126]. However, two patients with vascular sarcomas demonstrated prolonged disease stabilization of 10 months (hemangiopericytoma) and 19 months (epithelioid hemangioendothelioma) under Ang-(1-7) treatment, thereby requiring further investigation [126].”

  • Some sections are excessively information-dense with little background on the signaling pathways described, for example sections 3.1. Readers might not be familiar with all of the signaling pathways reported. Authors might want to consider simplifying some of these sentences to make the main concepts more easily understandable. Instead of a list of previously published evidence, the authors should convey more general concepts and then delve into some details.

Considering the reviewer suggestion, which we sincerely appreciated, we have simplified manuscript Figure S2, its legend, and the associated information inserted along the text. Furthermore, we also transformed Figure S2 into Figure 2.1 (Representation of potential angiotensin-associated pathways associated with cell proliferation, invasion, and migration in lung tumour microenvironment); Figure 2.2 (Potential angiotensin-associated pathways involved in hypoxia and angiogenesis in lung tumour microenvironment) and Figure 2.3 (Mechanistic representation of potential angiotensin-associated immunoinflammatory pathways in lung tumour microenvironment). We tried to simplify the text, however due to the complexity of the figures, we believe that any simplification to the text would reduce its accuracy.

  • Some sections are overly simplified. For example, in section 3.3 the authors describe the role of RAS components in inflammation and focus exclusively on macrophages. However, the effect of RAS on the immune systems (in particular myeloid cells) goes far beyond the modulation of the inflammatory response. For instance, RAS components are also expressed in other immune cells such as neutrophils and dendritic cells. A broader view of the role of RAS in the innate immune system would be desirable.

The authors tried to improve section 3.4 and also described the importance of neutrophils and dendritic cells in lung tumour microenvironment.

“3.4. Tumour Immunological Response

Recent cancer immunotherapies, including immune-checkpoint blockade, have produced durable clinical effects in some patients with various advanced cancers, namely in NSCLC [117]. Tumour anti-programmed death-ligand 1 (PD-L1) expression is associated with increased tumour-infiltrating lymphocytes in lung cancer. Unresponsiveness to the immune-checkpoint blockade therapies may be mediated by numerous immunosuppressive mechanisms that inhibit anti-tumour T-cell responses and T-cell infiltration into tumour tissues [118, 119]. In the lung cancer microenvironment, the cooperation between cancer cells namely airway epithelial cells, macrophages, and other peripherally‐recruited innate immune cells can determine the fate of lung tumours at different stages of both metastatic and primary disease, including preneoplasic, early, and late lesions (https://doi.org/10.1002/path.5241). Among the hematopoietic suppressor cells of interest are M2 macrophages, MDSC, and regulatory T cells (Treg), which have been associated with a poor prognosis in many cancers and with unfavourable clinical response to anti-PD-1/PD-L1 [118]. Furthermore, cancer-associated fibroblasts (CAFs) can manipulate the immune system, directly by inhibiting T and NK (natural killer) cell functions, promoting accumulation of suppressive cell types, and maintaining a pro-tumorigenic milieu. CAFs also induce tumour fibrosis that represents a physical barrier to T cell infiltration [120]. Neutrophils make up a significant portion of the inflammatory cell infiltrate in cancer, whereby they show high functional plasticity and display both anti-tumour and pro-tumour activities (https://doi.org/10.3892/ijo.2016.3616). Studies in early NSCLC have shown that neutrophils are pivotal to tumour cell clearance by stimulating adaptive immunity. In contrast, the role of neutrophils during later stages of primary lung cancer progression frequently appears to be pro‐tumorigenic, as neutrophils represent the most abundant recruited cells in more advanced NSCLC (https://doi.org/10.1002/path.5241). Tumour-derived signals induce a pro-tumour phenotype in neutrophils, which supports cancer cell invasion and metastasis (N2 neutrophils). N2 polarized neutrophils promote the proliferation, migration, and invasion of tumour cells, stimulate angiogenesis, as well as mediate immunosuppression (https://doi.org/10.3892/ijo.2016.3616). Dendritic cells are crucial for the activation of antigen-specific CD4 T lymphocytes, a pivotal step in the initiation of the innate and adaptive immune responses, which are essential for lung tumour cell clearance (https://doi.org/10.1186/s40880-019-0387-3). Previous studies demonstrated that lung cancers dynamically exclude functional DCs from the tumour region to support malignant progression. Furthermore, clinical trials have proven that dendritic cells function is reduced in lung cancer patients (https://doi.org/10.1186/s40880-019-0387-3). At the tumour microenvironment, the major components of RAS are also expressed both by resident and infiltrating cells, such as endothelial cells, fibroblasts, monocytes, macrophages, neutrophils, dendritic and T cells [8]. RAS signaling in these cells can facilitate or hinder growth and dissemination and has been shown to affect cell proliferation, migration, invasion, metastasis, apoptosis, angiogenesis, cancer-associated inflammation, immunomodulation, and tumour fibrosis/desmoplasia (DOI: 10.1126/scitranslmed.aan5616).”

Line 323: “TGF-β suppresses the differentiation and function of T helper, CD8+ cells, Natural Killer cells, and tumour-associated neutrophils, tumour associated macrophages and myeloid-derived suppressor cells. Tumour supporting cytokines are released from tumour and stromal cells upon AT1 receptor activation via Ang II including, TGF-β and Interleukins (IL-1a, IL-1b, IL-6, IL-8). Immunomodulatory cytokines may up-regulate immunosuppressive pathways, i.e. COX-2 via prostaglandin E2 synthesis, and impair dendritic cell function by reducing their migration. Ang II/AT1R signalling induces reactive oxygen species generation and related proteins such as inducible nitric oxide synthase in tumour and stroma cells. Exposure to ROS in the tumour microenvironment can impair T cell function while enhancing T regs and tumour associated macrophages. Treatment with the candesartan (AT1 receptor blocker) diminishes ROS generation (doi: 10.18632/oncotarget.26174). AngII/AT1R signalling is also important for myeloid differentiation and functional maturation. Cultured bone marrow from ACE 10/10 mice, a mouse line overexpressing ACE in monocytic cells, demonstrates increased myeloid maturation and reduced MDSC production; macrophages from these mice have an enhanced proinflammatory phenotype and antitumor activity compared to those from wild-type mice (DOI: 10.1038/labinvest.2014.41). Similarly, tumour-bearing ACE 10/10 mice showed enhanced immune response, which ultimately resulted in a reduced tumour growth. ACE inhibitors reversed the beneficial effects on tumour growth, but AT1 receptor blockade did not, suggesting that the effects of ACE overexpression were not dependent of AngII/AT1 receptor signalling (doi: 10.1038/nrneph.2018.15).”

  • Despite being beautifully comprehensive and designed, Figure S2 is far more detailed than the text referring to it. A simpler figure summarizing the main concepts described in the review might be more powerful at conveying the intended message.

Considering the reviewer suggestion, which we sincerely appreciated, we have simplified manuscript Figure 2, its legend, and the associated information inserted along the text. Furthermore, we also transformed Figure 2 into Figure 2.1 (Representation of potential angiotensin-associated pathways associated with cell proliferation, invasion, and migration in lung tumour microenvironment); Figure 2.2 (Potential angiotensin-associated pathways involved in hypoxia and angiogenesis in lung tumour microenvironment) and Figure 2.3 (Mechanistic representation of potential angiotensin-associated immunoinflammatory pathways in lung tumour microenvironment).

Specific comments 

Scientific comments (Major comments are specified):

  • MAJOR - Line 2 (Title) – The authors might consider revising the current title. “lung tumor and microenvironment interaction” is not clear. Also, throughout the review the authors refer to the “renin-angiotensin system” instead of “renin-angiotensin axis” as in the title and abstract. Please consolidate for consistency.

We sincerely appreciate the reviewer comment. As requested and for consistency, we modified throughout the entire document the “renin-angiotensin axis” to “renin-angiotensin system” and all modifications were highlighted in yellow. Exceptions correspond when within the renin-angiotensin system, we have exploited specific ligand/receptors axis as: the ACE/Ang II/AT1 axis, the ACE2/Ang-(1–7)/Mas or the AngII/ATR1 axis. Since we described the interaction of RAS and microenvironment, the authors suggest the following title: “Renin–angiotensin system in lung tumour and its microenvironment interactions”.

  • Line 18 and 34 – Clarify the difference between RAA (in the abstract) and RAS (throughout the text) and, if synonym, use one or the other definition throughout the text for consistency.

In fact, one can consider both term as synonyms, therefore, as explained in the response to the previous comment, we have consistently modified throughout the whole text the “renin-angiotensin axis” to “renin-angiotensin system” and all modifications were highlighted in yellow.

  • Line 21 – Please change/remove “In light of compelling evidence”.

Accordingly, we have removed “In light of compelling evidence” by “Angiotensin peptides are described to”.

  • Line 24-25 – Consider rephrasing “clinical intervention of this pathway”  to “blockade of this pathway in clinical settings”.

We appreciate the reviewer comment and have changed accordingly the sentence.

  • MAJOR - Line 43-44 – Here the authors should explain the physiological RAS hormone system/signaling much more into details, instead of referencing to previous reviews on the topic, in order to make the rest of the review more easily understandable to non-expert readers. In particular, the physiological role of angiotensin peptides should be explained here.

The authors tried to provide an overview of the normal function of the system and reformulated section 2 as previously described. Line 43-44: “The RAS was first discovered and studied in the physiological regulation of blood pressure, fluid homeostasis and pathogenesis of hypertension (1). Angiotensin II (AngII), the final effector of the system, causes vasoconstriction both directly and indirectly by stimulating Ang II type 1 receptor (AT1 receptor) present on the vasculature and by increasing sympathetic tone and arginine vasopressin release. Chronically, Ang II regulates blood pressure by modulating renal sodium and water reabsorption directly, by stimulating AT1 receptors in the kidney, or indirectly, by stimulating the production and release of aldosterone from the adrenal glands, or stimulating the sensation of thirst in the central nervous system (https://doi.org/10.1210/en.2003-0150). The enzymatic cascade by which Ang II is produced consists of renin, which cleaves angiotensinogen (AGT) to form the decapeptide angiotensin I. Ang I is then further cleaved by angiotensin-converting enzyme (ACE) to produce the octapeptide Ang II, the physiologically active component of the system. Further degradation by aminopeptidase A and N produces angiotensin III (Ang 2–8), and angiotensin IV (Ang 3–8), respectively (Figure 1) (https://doi.org/10.1210/en.2003-0150). The actions of Ang II result from its binding to specific receptors (AT1 and AT2). Ang II, via its AT1 receptor, is also involved in cell proliferation, left ventricular hypertrophy, nephrosclerosis, vascular media hypertrophy, endothelial dysfunction, neointima formation and processes leading to athero-thrombosis (DOI: 10.1080/08037050310001057). Ang II, via its AT2 receptor, is involved in cell differentiation, tissue repair, apoptosis and induces vasodilation (DOI: 10.1080/08037050310001057). Other receptors have been described in relation to the RAS. For instance, an AT4 receptor binding site has been identified, and in contrast to the other AT receptors does not seem to be a G protein-coupled receptor (doi: 10.1016/0167-0115(92)90527-2). This receptor binds Ang IV (3–8) preferentially, has been localized in various mammalian tissues, and has been suggested to cause vasodilatation (doi: 10.1016/0167-0115(92)90527-2).”

  • MAJOR - Lines 53-55 – Here, the authors make the point that intra-pulmonary RAS is involved in the etiology of lung diseases. I feel that this point should be made clearer/stronger, followed by the list of experimental evidence that the authors nicely summarize in lines 57-116.

As suggested by the reviewer, we have improved our introduction concerning the importance of intra-pulmonary RAS in the development of lung diseases.

Line 54: “Alterations in RAS expression were shown to be involved in multiple lung diseases: idiophatic pulmonary fibrosis (DOI: 10.1183/09031936.00130310), sarcoidosis (https://doi.org/10.1177/1470320313489059), pulmonary hypertension (DOI: 10.1164/rccm.201203-0411OC), acute respiratory distress syndrome (DOI: 10.1016/j.pupt.2019.101833), lung cancer (DOI: 10.21873/anticanres.13639). Understanding the involvement of intra-pulmonary RAS in inflammation or fibrosis may open new therapeutic possibilities for the treatment of respiratory diseases.”

  • Line 77-78 – AGTR1 and AGTR2 should be defined here. What proteins do they encode?

We have now, as suggested, introduced the name of the proteins encoded by the AGTR1 and AGTR2 genes.

  • Line 85 – Transition of what cells?

We appreciate the comment of the reviewer and substituted “the transition to the myofibroblast phenotype” to “the epithelial to mesenchymal transition” to clarify.

  • Line 88 – What does local refer to? I guess lungs, but please explain.

In this sentence we rather meant local in opposition to systemic regulation. In the context of this sentence, local upregulation is concerned to the lungs but in other contexts it may be organ specific. For this reason, we believe that it could be more correct to leave it as “local upregulation” but we acknowledge the review question.

  • MAJOR - Lines 43-116 (section 2) – In section 3, the authors describe that a variety of pro- or anti-tumor effects of angiotensin peptides rely on their signaling through AT1 and AT2 on tumor cells. In section 2 or a separate section the authors should focus on the expression pattern of AT1 and AT2 receptor in cancer cell lines, patient biopsies and tumor microenvironment of the lungs. Some of these pieces of information are scattered throughout the text and should be combined in the same section. A paragraph on the known mutations affecting these factors in cancer would also be very informative.

The authors would like to thank you for pointing this out. We reformulated section 2 and explored the distribution of RAS components in lung adenocarcinoma as described above. Moreover, the authors also explored the role of the circulating RAS deregulation in lung diseases including lung cancer and described the known germline mutations that might affect RAS components.

Line 80: “The components of RAS are frequently differentially expressed in various cancers including brain, lung, pancreatic, breast, prostate, colon, skin and cervical carcinomas in comparison with their corresponding non-malignant tissue (doi:10.1093/carcin/bgn171). A large-scale study of estrogen receptor-positive breast cancer tumours revealed an increase in AGTR1 mRNA expression. Contradictory results were observed in lung adenocarcinoma tissues since AGTR1, AGTR2 as well as ACE mRNA expression were underexpressed. Notably, AGT was overexpressed in lung adenocarcinoma tissue. (PMID 28791183, https://doi.org/10.1371/journal.pone.0226553).” 

Line 114: “The circulating RAS might also be important in providing the substrates for local modulation of angiotensin peptides with influence in lung diseases. Genetic and epidemiological evidence provides support that germline mutations in RAS components contribute to the risk of developing idiopathic pulmonary fibrosis, acute lung injury and certain malignancies. For example, the human gene encoding angiotensinogen (AGT) is located on chromosome 1q42.3 and a large number of single-nucleotide polymorphisms (SNPs) have been described. Among them, the SNPG-6A nucleotide substitution at position -6 upstream from the initial transcription start has been studied particularly. The A allele has been associated in vitro with an increased expression of the AGT gene and with higher AGT synthesis. A previous report showed that the G-6A polymorphism of the AGT gene is associated with increased angiotensin production and idiopathic pulmonary fibrosis progression [DOI:10.1183/09031936.00015808]. In the limited number of studies that have been carried out, the AGT M235T variant conferred a risk for developing breast cancer in post-menopausal women. Moreover, ACE plasma levels depend on a 287 bp insertion/deletion (I/D) polymorphism of the ACE gene located on chromosome 17q23. Homozygotes for the D allele have the highest ACE plasma levels, homozygotes for the I allele have the lowest, and ID heterozygotes have intermediate levels. Several studies found an association between ACE I/D polymorphism and lung cancer, albeit some conflicting results [doi:10.1177/1470320314552310]”.

  • Line 118 – RAS is not a mechanism but a signaling system. Please rephrase.

Accordingly to the reviewer request, we changed the expression “The RAS is an important mechanism” to “The RAS is an important system”.

  • Line 120 – “Metabolism” is not a tumor-associated phenotype, please rephrase.

In agreement with the reviewer comment, we changed the expression “in other tumour-associated phenotypes, such as angiogenesis, metabolism, migration” to “in the regulation of metabolic homeostasis and of tumour-associated phenotypes, such as angiogenesis, migration and invasion”.

  • Line 121 and section 3.1. - Tumor cell proliferation and metastasis/invasion/migration are two distinct processes in cancer progression. If possible, the authors should attempt to address these two aspects in separate subsections of chapter 3.

We absolutely agree with the reviewer comments. During the preparation of the manuscript we had exactly the same concern, nevertheless, considering that many of the in vitro studies simultaneously evaluated the impact of RAS peptides, receptors or inhibitors on cancer cell proliferation, migration and invasion, for sake of simplicity, we decided to maintain this distinct cellular processes within the same section.

  • Line 144-146.- This sentence seems to be unrelated to the previous one. Consider moving this somewhere else.

We appreciate the reviewer comment and have now reformulated the beginning of the sentence to integrate it with the previous and the subsequent sentences. Accordingly we state that “Notably, the impact of AngII/AT1 receptor axis on cell growth and proliferation has been also accompanied by evidences of impairment of apoptotic pathways. Accordingly, Telmisartan, an AT1 receptor antagonist, significantly inhibited the growth of non-small cell lung cancer (NSCLC) A549 cell line in a time- and dose-dependent manner and led to an increase of pro-apoptotic proteins caspase-3 and Bcl-xL [54].”

  • Line 149 – “Neo-vessel” is not a conventional scientific terminology. Please change to “new vessels” or “neovascularization”.

We have changed the term “neo-vessel formation” by “neovascularization”, as suggested by the reviewer.

  • Line 150 – The role of CSC in cancer metastasis is still a matter of debate and the mechanisms are not yet completely uncovered. The authors should rephrase the expression”compelling evidence demonstrating...” to make it less decisive.

As proposed, we have changed the expression “compelling evidence demonstrating” by “some studies suggest”.

  • Line 151-155 – Although ang II/ATR have a role in EMT, EMT was not listed among mechanisms of cancer metastasis in the previous sentence. Please expand.

We appreciate the reviewer comment and had therefore expanded the previous sentence as: “Tumour metastasis involves several biological steps, such as cell-to-cell and cell-to-extracellular matrix (ECM) interactions, epithelial-to-mesenchymal transition (EMT), degradation of ECM components, neovascularization [55]…”

  • Line 157 – Does AGTR1 refer to the gene (AGTR1, in italic) or the protein (ATR1)?

AGTR1 was referring to the ATR1 receptor and has, therefore, been changed accordingly.

  • Line 158 – A lot of the abbreviations in this sentence (CXCR4/SDF-1α signaling through  FAK/RhoA/ROCK1/2/MLC) should be expanded and/or omitted for clarity.

We appreciate the reviewer detailed observation and had expanded the different abbreviations for clarity as: “by upregulating the C-X-C chemokine receptor type 4 (CXCR4)/Stromal cell derived factor-1α (SDF-1α) signalling through the Focal adhesion kinase (FAK)/Ras homolog family member A (RhoA)/Rho associated kinase 1/2 (ROCK1/2)/Myosin light-chain kinase (MLC) pathway”.

  • MAJOR - Line 123-143 – Despite the sentence on lines 159-161, the pro-proliferative and pro-metastatic roles of Angiotensin II/ATR1 are general and, in most cases, do not appear to have been demonstrated in lung cancer models, either in vitro or in vivo. As highlighted in figure 2, these pathways might “potentially” apply to lung cancer. The authors should highlight the relevance of these mechanisms to lung cancer, if any. Similar considerations need to be made for section 3.1.2 and 3.1.3.

Thank you for pointing this out. Although we agree that this is an important consideration, the authors included in vitro studies that demonstrate the role of RAS in lung cancer proliferation/invasion (lines 147, 156, 158).

The remaining cancer models were also described in order to highlight the different signalling pathways that contribute for cancer cell proliferation, migration and invasion. Furthermore, we described ERK/p38/MAPK pathway and TGFb1/Smad2/Smad3/Smad4 signalling pathway for a better understanding of figure 2.1. Line 143: “Notably, in fibroblasts, Ang II binding to AT1R results in Smad2 and Smad3 phosphorylation via the ERK/p38/MAPK pathway, increasing the activation of the TGFb1/Smad2/Smad3/Smad4 signalling pathway [DOI: 10.1186/s13069-015-0023-z].” 

Section 3.1.3: Regarding NFAT, the studies in cardiomyocytes have not been found in lung or tumour. Nevertheless, accumulating studies have suggested that NFATs are involved in many aspects of cancer, including carcinogenesis, cancer cell proliferation, metastasis, drug resistance and tumour microenvironment [https://doi.org/10.1016/j.canlet.2015.03.005].

In fact, the crosstalk between Ang-(1-7) and NFAT in lung cancer has never been explored. As such, the authors reformulated the statement: “Further data from cardiomyocytes supports a nitric oxide/guanosine 3’,5’-cyclic monophosphate-dependent pathway, which modulated the activity of the transcription factor NFAT (nuclear factor of activated T-cells), preventing its translocation to the nucleus [73], where is known to transcriptional regulate genes involved in malignant cell proliferation, migration and metastasis [74]. However, the crosstalk between ACE2/Ang-(1-7)/Mas axis and NFAT upregulation in lung cancer has never been explored and the relationship to tumours is speculative based on studies in other tissues.”

  • Line 196-197 – Evidence presented by the authors supports the idea that increased ACE2 activity and Ang(1-7) levels have anti-tumor and anti-metastasis effects. In this sense “targeting” should be rephrased to emphasize that the overexpression, and not the blockade, of these proteins might be beneficial in the context of lung cancer. Comments should also be made in regards to what strategy might be more effective to induce the activation of the ACE2/Ang(1-7)/Mas axis in patients, if any.

We appreciated the reviewer comment. The authors improved section 4.1 (ClinicalTrials) where we described a small phase I study about the administration of Ang-(1-7) in cancer patients. Clinical benefit, defined as disease stabilization for more than three months, was observed in two of the three patients with metastatic sarcoma. The activation of ACE2/Ang-(1-7)/Mas axis could be achieved by the administration of Ang-(1-7) or ACE2 activators, exogenously or through nanoparticles. However, due to the speculative nature of this statement, the authors opted not to include it in the manuscript.

  • MAJOR - Line 198 -  A notable omission of the authors is the paper: Attoub S, Gaben AM, Al-Salam S, et al. Captopril as a potential inhibitor of lung tumor growth and metastasis. Ann N Y Acad Sci. 2008;1138:65‐ doi:10.1196/annals.1414.011, which provides pre-clinical data of the effect of captopril on lung tumor growth and metastasis.

As suggested by the reviewer, we have added this relevant study in order to highlight the role of inhibitors of ACE in lung tumour growth. Line 146: “Attoub S et al., 2008 demonstrated that captopril significantly inhibited tumour growth using a non-small cell lung carcinoma LNM35 preclinical model”.

  • Lines 200-207 – This paragraph should introduce the subsections 3.2.1-2. Experimental data regarding the role of AngII, ACE and AT1 in hypoxia and angiogenesis should be moved to the following subparagraphs.

We appreciated your comment. However considering the 3.2 section is devoted to hypoxia we believe that this experimental data is propriety located within this subsections.

Line 206 – RAS is a system and it is not “expressed”. Please rephrase.

Accordingly, we have rephrased to “RAS proteins expression”.

  • Line 214 – Expand on candesartan pharmacological action (e.g. target?), for clarity. Non-experts are not familiar with this drug.

We appreciate this observation and supplied information on candesartan action by adding. “… following candesartan administration, an angiotensin receptor inhibitor”.

  • Line 215 and 218-219 – “...mRNA and protein...” and “...at both mRNA and protein level” are details not needed here. “Expression” should suffice.

As suggested, the word “expression” was added to the statement.

  • MAJOR - Line 236 – It is clear from other reviews that the effect of RAS components on the immune system goes beyond modulation of the inflammatory response. RAS components are also expressed in neutrophils and dendritic cells. What is the role of these myeloid cells in lung cancer etiology and how does the RAS affect it? Authors should expand on the role of RAS in the innate immune system of lung cancer beyond inflammation.

We agree with the reviewer’s assessment. However, there is sparse information considering the role of RAS in the innate immune system of lung cancer. The authors tried to improve section 3.4 and also described the importance of neutrophils and dendritic cells in lung tumour microenvironment as described above.

“3.4. Tumour Immunological Response

Recent cancer immunotherapies, including immune-checkpoint blockade, have produced durable clinical effects in some patients with various advanced cancers, namely in NSCLC [117]. Tumour anti-programmed death-ligand 1 (PD-L1) expression is associated with increased tumour-infiltrating lymphocytes in lung cancer. Unresponsiveness to the immune-checkpoint blockade therapies may be mediated by numerous immunosuppressive mechanisms that inhibit anti-tumour T-cell responses and T-cell infiltration into tumour tissues [118, 119]. In lung cancer microenvironment, the cooperation between cancer cells namely airway epithelial cells, macrophages, and other peripherally‐recruited innate immune cells can determine the fate of lung tumours at different stages of both metastatic and primary disease, including preneoplasic, early, and late lesions (https://doi.org/10.1002/path.5241). Among the hematopoietic suppressor cells of interest are M2 macrophages, MDSC, and regulatory T cell (Treg), which have been associated with a poor prognosis in many cancers and with unfavourable clinical response to anti-PD-1/PD-L1 [118]. Furthermore, cancer-associated fibroblasts (CAFs) can manipulate the immune system, directly by inhibiting T and NK (natural killer) cell functions, promoting accumulation of suppressive cell types, and maintaining an inflammatory pro-tumorigenic milieu. CAFs also induce tumour fibrosis that represents a physical barrier to T cell infiltration [120]. Neutrophils make up a significant portion of the inflammatory cell infiltrate in cancer, whereby they show high functional plasticity and display both antitumour and pro-tumour activities (https://doi.org/10.3892/ijo.2016.3616). Studies in early NSCLC have shown that neutrophils are pivotal to tumour cell clearance by stimulating adaptive immunity. In contrast, the role of neutrophils during later stages of primary lung cancer progression frequently appears to be pro‐tumorigenic, as neutrophils represent the most abundant recruited cell type in more advanced NSCLC (https://doi.org/10.1002/path.5241). Tumour derived signals induces a pro-tumour phenotype in neutrophils, which supports tumour growth and metastasis (N2 neutrophils). N2 polarized neutrophils promote the proliferation, migration, and invasion of tumour cells, stimulate angiogenesis, as well as mediate immunosuppression (https://doi.org/10.3892/ijo.2016.3616). Dendritic cells are crucial for the activation of antigen-specific CD8 T lymphocytes, a pivotal step in the initiation of the innate and adaptive immune responses, which are essential for lung tumour cell clearance (https://doi.org/10.1186/s40880-019-0387-3). Previous studies demonstrated that lung cancers dynamically exclude functional DCs from the tumour region to support malignant progression. Furthermore, clinical trials have proven that dendritic cells function is reduced in lung cancer patients (https://doi.org/10.1186/s40880-019-0387-3). At the tumour microenvironment, the major components of RAS are also expressed both by resident and infiltrating cells, such as endothelial cells, fibroblasts, monocytes, macrophages, neutrophils, dendritic cells, and T cells [8]. RAS signaling in these cells can facilitate or hinder growth and dissemination and has been shown to affect cell proliferation, migration, invasion, metastasis, apoptosis, angiogenesis, cancer-associated inflammation, immunomodulation, and tumour fibrosis/desmoplasia (DOI: 10.1126/scitranslmed.aan5616).”

Line 323: “TGF-β suppresses the differentiation and function of T helper, CD8+ cells, Natural Killer cells, and tumour-associated neutrophils, tumour associated macrophages and myeloid-derived suppressor cells. Tumour supporting cytokines are released from tumour and stromal cells upon AT1 receptor activation via Ang II including, TGF-β Interleukins (IL-1a, IL-1B, IL-6, IL-8). Immunomodulatory cytokines may up-regulate immunosuppressive pathways, i.e. COX-2 via prostaglandin E2 synthesis, and impair dendritic cell function by reducing their migration. Ang II/AT1R signalling induces reactive oxygen species generation and related proteins such as inducible nitric oxide synthase in the tumor cells and stroma cell. Exposure to ROS in the tumour microenvironment can impair T cell function while enhancing T regs and tumour associated macrophage. Treatment with the candesartan (AT1 receptor bloacker) diminishes ROS generation (doi: 10.18632/oncotarget.26174). AngII/AT1R signalling is also important for myeloid differentiation and functional maturation. Cultured bone marrow from ACE 10/10 mice, a mouse line overexpressing ACE in monocytic cells, demonstrates enhanced myeloid maturation and reduced MDSC production; macrophages from these mice have a more proinflammatory phenotype and more antitumor activity compared to those from wild-type mice (DOI: 10.1038/labinvest.2014.41). Similarly, tumour-bearing ACE 10/10 mice showed enhanced immune response, which ultimately resulted in a reduced tumour growth. ACE inhibitors reversed the beneficial effects on tumour growth, but AT1 receptor blockade did not, suggesting that the effects of ACE overexpression were not dependent of AngII/AT1 receptor signalling (doi: 10.1038/nrneph.2018.15).”

  • Line 290 – Inflammation is a process and cannot be upregulated. Please rephrase.

We appreciate the reviewer correction and reformulated the sentence as “…thus controlling ROS production and subsequent inflammation”.

  • MAJOR - Line 306-326 – Similarly to line 123-143, the studies reported here were done in cancer models different to lung cancers. Authors should address the relevance of this pathways to lung cancer, if any. Also, for consistency with the other subsection of section 3, also this paragraph should be formatted as “ACE/Ang II/AT1 receptor axis and AT2 receptor” and “ACE2/Ang(1-7)/Mas receptor axis”, if possible.

Thank you for pointing this out. Some reports suggest that targeting RAS could alleviate immunosuppression and enhance the outcome of immunotherapy. However, these studies were performed in other cancer models. Hence, “it is necessary to get into further detailed knowledge regarding RAS through experimental and clinical research in lung cancer and its influence in immunotherapy response” (Line 323-326). The authors also followed the reviewer´s suggestion and changed to “ACE/Ang II/AT1 receptor axis and AT2 receptor” and “ACE2/Ang(1-7)/Mas receptor axis”.

  • MAJOR - Line 333 and section 4 – A major omission of the authors is the paper “Aydiner, Adnan MD; Ciftci, Rumeysa MD; Sen, Fatma MD Renin-Angiotensin System Blockers May Prolong Survival of Metastatic Non-Small Cell Lung Cancer Patients Receiving Erlotinib, Medicine: June 2015 - Volume 94 - Issue 22 - p e887 doi:10.1097/MD.0000000000000887”, where RAS blockers in combination with erlotinib improve overall survival of NSCLC patients.

As suggested by the reviewer, we have added this relevant study.

Line 349: “A retrospective study with 117 metastatic NSCLC patients demonstrated that the intake of AT1 receptor blockers, during erlotinib treatment, may prolong overall survival”.

  • MAJOR - Lines 334-343 – Considering the lack of clinical trials on lung cancer, which is the main focus of this review, I would consider omitting the subsection 4.1, unify subsection 4.2 with 4 and add the sentences of lines 334-337 to the newly unifies section.

Although there is a lack of clinical trials, the authors explored further this issue and improved this section, describing the most relevant clinical trials and respective results. Therefore, the authors reformulated section 4.1: “Despite basic and pre-clinical evidence concerning the impact of RAS in cancer hallmarks, the information available regarding clinical trials remains scarce, particularly in lung cancer. The primary objective of a Phase II Randomized Trial (NCT00077064, table S1) was to test the ability of captopril to alter the incidence of pulmonary damage at 12 months after completion of radiation in patients with non-small cell lung cancer or limited-stage small cell lung cancer. Due to the unmet accrual/randomization goals, it was not possible to adequately test the hypothesis that captopril ameliorates radiotherapy-induced pulmonary toxicity. Another double-blind placebo-controlled randomized trial (NCT01880528, table S1) enrolled 23 patients with lung cancer to study the putative protective effect of lisinopril in pneumonitis induced by radiotherapy. The results of this clinical trial suggest that there was a clinical signal for safety and possibly beneficial in concurrently administering lisinopril during thoracic radiotherapy to mitigate or prevent radiation-induced pulmonary distress [doi: 10.1016/j.ijrobp.2018.10.035]. However, larger-scale randomized phase 3 trials are needed in the future to confirm these results. Notably, a multicenter clinical trial showed that losartan stabilized lung function in patients with idiopathic pulmonary fibrosis over 12 months (NCT00879879, table S1), reinforcing the importance of AT1 receptor blockers in the impairment of fibrosis progression. Moreover, the treatment with AT1 receptor blockers might potentiate the response to immunotherapy in lung cancer patients due to a hypothetic impairment of a fibrotic immunosuppressive microenvironment. Nevertheless, there is no study to prove this hypothesis. A small Phase I study (NCT00471562, table S1) of Ang-(1–7) as a first-in-class anti-angiogenic drug targeting Mas receptor, was undertaken in 18 patients with advanced solid tumours refractory to standard therapy with only one patient with lung cancer included [125]. Dose-limiting toxicities encountered at the 700 μg/kg dose included stroke (grade 4) and reversible cranial neuropathy (grade 3). Other toxicities were generally mild. Clinical benefit, defined as disease stabilization for more than three months, was observed in two of the three patients with metastatic sarcoma and it was associated with reduction of PlGF plasma levels [125]. A Phase II study failed to confirm PlGF as biomarker of response to Ang-(1-7) administration [126]. However, two patients with vascular sarcomas demonstrated prolonged disease stabilization of 10 months (hemangiopericytoma) and 19 months (epithelioid hemangioendothelioma) under Ang-(1-7) treatment, thereby requiring further investigation [126].”

  • Line 352 – Please rephrase “adjuvant specific biological mechanism of action” for clarity.

As suggested by the reviewer, for clarity, we have reformulated: “suggesting the existence of an adjuvant effect for angiotensin blockers.

  • Line 356-357 – Is it an increased or decreased risk of lung cancer that was not confirmed in patients under ARBs? Please correct sentence.

This systematic review concludes that Angiotensin receptor blockers (ARBs) are indeed associated with a reduced lung cancer risk (OR=0.81, 95% CI 0.69-0.54).

  • MAJOR - Line 371-374 – The sentence “The major challenge in clinical context is the complex nature of RAS signalling, which seems to be variable in response to several factors, making the response to RAS therapeutics, either individually or in combination with other drugs, difficult to predict” is particular important for the purpose of this review. I suggest that the authors overview these aspects into details under the section “Clinical trials”.

The authors aprreciated your comment. Despite the difficulty in performing clinical trials, the authors improved section 4.1. See the comments above.

Figure S2 – This figure has far more molecular details that the text referring to it (sections 3). Figure should be simplified to reflect the information that is written in the main body of the review.

Considering the reviewer suggestion, which we sincerely appreciated, we have simplified manuscript Figure 2, its legend, and the associated information inserted along the text. We hope that now you can find it more comprehensible. Furthermore, we also transformed Figure 2 into Figure 2.1 (Representation of potential angiotensin-associated pathways associated with cell proliferation, invasion, and migration in lung tumour microenvironment); Figure 2.2 (Potential angiotensin-associated pathways involved in hypoxia and angiogenesis in lung tumour microenvironment) and Figure 2.3 (Mechanistic representation of potential angiotensin-associated immunoinflammatory pathways in lung tumour microenvironment)

  • Figures S1 and S2 – Both figures should be moved to the main text, upon simplification (Figure 2) and shortening of the figure legends.

Considering the reviewer suggestion, which we sincerely appreciated, we have simplified manuscits legend, and the associated information inserted along the text. We also included the figures in the main text.

Language comments:

  • Lines 18-25 (Abstract) and all text - Consider rephrasing the expressions “supporting/recent/compelling evidence” or “previous studies” with other forms (e.g. first author of cited paper) to avoid repetition or removing them to make reading more fluid. In particular, the abstract should be reviewed to avoid repetition of the word “evidence”.

We appreciate this comment and had reformulated some expressions along the abstract and the rest of the manuscript, avoiding words repetition to improve reading.

  • Lines 29-31 – Review sentence to avoid repetition of “worldwide”.

Accordingly, the 1st “worldwide” was removed from the sentence. We appreciate your attentive work.

  • Line 34 - The abbreviation RAS for “renin-angiotensin-aldosterone system” should be defined here, where it appears for the first time.

Accordingly, we have introduced the abbreviation “RAS” to explain the expression “renin-angiotensin-aldosterone system”.

  • Line 35, 222 and 370 – “Notwithstanding” is not commonly used in English, consider changing to “Nevertheless”.

We have now substituted the expression “Notwithstanding” by the expression “Nevertheless”, as suggested. Thank you for your comment!

  • Line 37 – “,thereby critically” is incorrect here, please rephrase.

As suggested, we rephrased the sentence to “actively participating in lung tumour microenvironment regulation”.

  • Line 38 – Please change “updated information”.

We rephrased the sentence to “..we review current knowledge on the renin-angiotensin system (RAS)...”.

  • Line 44-45 – The definition of RAS should be introduced before and only the abbreviation used here.

We appreciate the comment and corrected accordingly.

  • Line 46-49 – consider rephrasing to avoid repetition of “local”.

The sentence was now rephrased to “In addition, supportive evidence claims that tissue RAS is capable of working synergistically or independently of the systemic RAS, and locally generates angiotensin peptides with regulatory homeostatic functions, thus contributing to dysfunction and/or disease…”.

  • Line 51 – “On its turn” is not a correct English expression and should be changed to “In turn” or “subsequently”, although the exact connector meant by the authors is not clear here.

We appreciate the comment and have now changed the expression “On its turn…” to “In turn…”.

  • Line 53 – “In fact” is not used correctly here, as this sentence does not explain the previous one. Please rephrase or remove.

As suggested, the word “In fact” was removed from the beginning of the sentence.

  • Line 55-56 – Rephrase “Add to knowledge supporting...”.

Considering reviewer suggestion, the expression “…add to knowledge supporting” was substituted by “…supports the existence”.

  • Line 74, 94, 122, 125, 130, 141, 151 and throughout the text– “Ang” should be expanded to angiotensin for consistency or, alternatively, angiotensin should be abbreviated to “ang” throughout the text.

We appreciate very much the comment and have changed the “Angiotensin” terms by “Ang” for consistency.

  • Line 89-90 – Please rephrase “which might influence each other’s activity or synergy” for clarity.

For clarity, the word “synergy” was removed from the sentence.

  • Line 114 – “having as surrogate the hallmarks of cancer” is incorrect, please revise.

The authors reformulated the statement: “Furthermore, some studies reported that the ACE2/Ang-(1–7)/Mas axis counteracted the ACE/Ang II/AT1 axis in different models of cancer, including lung cancer”.

  • Line 126-130 – Consider dividing this sentence in 2 shorter ones.

We appreciate the reviewer suggestion and have therefore divided the sentence into two as: “…members. This crosstalk was shown to contribute towards cancer [5] (Figure S2)”.

  • Line 136 – Change “is incompletely understood” to “is not completely understood”. “Support a role ... as synergistically efficacious” is not correct, please rephrase to “show an effect” or similar.

We have modified the expressions consistently with the reviewer’s suggestions.

  • Line 142  - Rephrase “all intracellular signaling pathways”.

We have rephrased the expression, as requested.

  • Line 147 – Change “related” with “associated” or similar.

We have modified the expression, as requested.

  • Line 171 – “...among..” is used incorrectly here. Please correct. “Signaling” is US English. Considering that the rest of the review has been formatted in UK English, please change to “signalling”.

As recommended, the word “among” was changed to “of”. This and other words that were still in US English were corrected to UK English.

  • Line 178-179 – “Some studies demonstrated that...” and “...have been associated with” are incompatible in the same sentence. Please rephrase.

We have rephrased the sentence in a correct manner.

  • Line 183 – The abbreviation “COX” needs to be expanded to “cyclooxygenase (COX)...”.

The abbreviation was expanded to its complete designation. We appreciate the suggestion.

  • Line 184-185 – “By an effect to under-express...” is incorrect. Please rephrase. “Under-express” and “under-activate” are not conventional scientific language, please revise.

The authors reformulated: “It was demonstrated that Ang-(1-7) reduced lung cancer cell migratory and invasive abilities through the decrease of expression and activity of matrix metalloproteinases MMP-2 and MMP-9 and inactivation of the PI3K/Akt, P38 and JNK signalling pathways”

  • Line 189 – “in A breast cancer model”.

The suggested correction was performed.

  • Line 191 – Correct “e-cadherin” to “E-cadherin”

The suggested correction was performed.

  • Line 194 – “where is known to transcriptional regulate” is incorrect and needs rephrasing.

The expression was rephrased to “…and impairing the transcriptional regulation of genes…”.

  • Line 205 – “...previous studies demonstrated that...” is not needed here.

The expression was removed.

  • Line 220 – Reference  [84] should be moved to the end of the sentence.

The reference was, as suggested, removed to the end of the sentence.

  • Line 222 – “Notwithstanding” is not correct here, as there is no connection with the previous sentence. Consider changing to “In contrast to AT1...” or similar.

We appreciate the correction and have altered the expression accordingly the reviewer suggestion.

  • Line 238 – Consider changing “major precursors”.

The expression has been rephrased.

  • Line 239 – “...microenvironment characteristics” cannot be rich in something, but tumor microenvironment can. Please revise.

The sentence has been revised, as suggested.

  • Line 240 – “cell-proliferation” is pleonastic, please remove.

The expression has been removed.

  • Line 241 – Rephrase to “Inflammation has been postulated to play a key role in lung carcinogenesis” or similar.

The sentence has been modified as suggested.

  • Line 243 – Use either “et al.” or “and colleagues” throughout the text, for consistency.

We decided to use the expression “and colleagues” throughout the whole text.

  • Line 245-247 – The sentence “exposure of agents...inflammation” is incorrect. Please revise.

  • Line 251 – Change “RA” to “RAS”.

The alteration has been performed.

  • Line 253 – “Intensely” is not correctly used here, please change.

The word “intensely” has been removed.

  • Line 264 – “blockage” should be substituted with “blockade”.

The substitution was made accordingly.

  • Line 270 – Please rephrase “Despite conflicting data, Ang II/AT1 receptor signalling stimulates...”.

The sentence was rephrased to “Although some controversy in the field, Ang II/AT1 receptor signalling seems to stimulate the expression of cytokines and growth factors…”

  • Line 273 –  “...albeit, taken together” is incorrect. Consider starting a new sentence and rephrase.

The expression “..albeit, taken together most studies” was changed to “…albeit most studies”.

  • Line 276 – “Conversely from” is incorrect, please change.

In opposite from AT1 receptor activation by Ang II, the Ang-(1-7) binding to Mas receptor triggers a down regulation of pro-oxidant pathways, preventing or attenuating the oxidative stress-induced cellular damage (98). In addition, Ang-(1-7) can also inhibit Ang II-activated inflammation through a deregulatory effect in lectin-like oxidized low-density lipoprotein receptor-1.

  • Line 279 – “cumulatively” is not used correctly here. Please remove.

The word has been removed.

  • Line 283 – Change “in macrophage function” to “on macrophage function”?

The expression was changed, as suggested.

  • Line 319 – Please correct “this result led to...”.

The expression “this result led” was changed to “this led”.

  • Line 325 – The phrase “get into further detailed knowledge regarding RAS through experimental and clinical research in lung cancer and its influence in immunotherapy response” needs rephrasing.

The sentence was changed to: “Nevertheless, it is necessary, through experimental and clinical research, to highlight the role of RAS in lung cancer and its influence on immunotherapy response.”

  • Line 339-340 – If PlGF is equal to PLGF previously indicated in the text, please format similarly.

The alteration was made consistently throughout the whole text.

  • Line 353 – “supported” is not correct here, please change to “showed” or similar.

The alteration was made.

  • Line 355 – A large cohort cannot “suggest”, please rephrase.

The sentence was rephrased to “, a study on a large population-cohort using iACE suggested an…”

  • Line 356 – Please correct “subjects with more than five years under treatment”.

The sentence was corrected to: “…patients undergoing treatment for more than five years…”.

  • Line 367 – Consistency is needed throughout the text for the terms “in vivo” and “in vitro” in terms of italics font. The use of “” here is not needed.

Alterations were performed throughout the whole text for consistency.

  • Line 377 – Please correct the typo “oppening”.

This and other typo mistakes have been corrected.

  • Line 379 and 382 – Consider inverting “Finally” and “Moreover”.

The alterations were performed accordingly.
